# Coupled atmosphere-ice-ocean dynamics during Heinrich Stadial 2

Xiyu Dong [1], Gayatri Kathayat[1] ✉, Sune O. Rasmussen [2], Anders Svensson [2], Jeffrey P. Severinghaus[3], Hanying Li[1], Ashish Sinha [1,4], Yao Xu[1], Haiwei Zhang [1], Zhengguo Shi[1,5,6], Yanjun Cai [1], Carlos Pérez-Mejías[1], Jonathan Baker [1], Jingyao Zhao[1], Christoph Spötl [7], Andrea Columbu [8], Youfeng Ning[1], Nicolás M. Stríkis [9], Shitao Chen[10,11,12], Xianfeng Wang [13], Anil K. Gupta[14], Som Dutt[15], Fan Zhang[1], Francisco W. Cruz[16], Zhisheng An [5], R. Lawrence Edwards[17] & Hai Cheng [1,5,18] ✉

Our understanding of climate dynamics during millennial-scale events is incomplete, partially due to the lack of their precise phase analyses under various boundary conditions. Here we present nine speleothem oxygen-isotope records from mid-to-low-latitude monsoon regimes with sub-centennial age precision and multi-annual resolution, spanning the Heinrich Stadial 2 (HS2) − a millennial-scale event that occurred at the Last Glacial Maximum. Our data suggests that the Greenland and Antarctic ice-core chronologies require +320- and +400-year adjustments, respectively, supported by extant volcanic evidence and radiocarbon ages. Our chronological framework shows a synchronous HS2 onset globally. Our records precisely characterize a centennial-scale abrupt "tropical atmospheric seesaw" super-imposed on the conventional "bipolar seesaw" at the beginning of HS2, implying a unique response/feedback from low-latitude hydroclimate. Together with our observation of an early South American monsoon shift at the HS2 termination, we suggest a more active role of low-latitude hydroclimate dynamics underlying millennial events than previously thought.

The last glacial period was characterized by a recurrence of millennial-scale cold-dry stadials and warm-humid interstadials, recognized from the Greenland ice-core records, known as Dansgaard-Oeschger oscillations[1,2]. Some Greenland Stadials have been associated with major iceberg discharge episodes and concomitant ice-rafted debris deposition in the central North Atlantic[2,3]. Previously marine studies have identified and characterized six such episodes known as Heinrich events during the last 60 thousand years[3,4]. The marine records show dramatic responses to Heinrich events for extended periods of time, often towards the end of the corresponding Greenland Stadial (e.g., ref. 5). We refer to these periods of extreme conditions as Heinrich Stadials (HSs), and note that they are not defined by the duration of the Greenland Stadial counterpart. Mid-to-low-latitude monsoon proxy records suggest a major weakening of the Asian summer monsoon (ASM) during the HSs[6,7] [hereafter Asian Heinrich Period (AHP)] and a strengthening of the South American summer monsoon (SASM)[8] [hereafter South American Heinrich Period (SAHP)]. The weakening and strengthening of the two monsoon systems, respectively, are associated with the southward displacement of the Intertropical Convergence Zone (ITCZ)[6]. An abrupt positive (negative) isotope excursion during the AHP (SAHP) is well-expressed in the speleothem oxygen-isotope ($\delta^{18}O$) records from the interhemispheric monsoon domains[8,9] (Supplementary Fig. 1c). It is noteworthy that Heinrich events have left a relatively small imprint in the Greenland ice-core $\delta^{18}O$ records[10].

Marine proxy records and numerical simulation experiments suggest that the Atlantic meridional overturning circulation (AMOC)

weakened or even shut down during the HSs[11]. Iceberg discharge and associated freshwater flux almost led to a stop of the deep-water formation in the North Atlantic, which in turn weakened the AMOC[11]. Hence, the northward oceanic heat transport was reduced, resulting in Greenland cooling and corresponding Antarctica warming with a time lag of $122 \pm 24$ years[12,13]; this phenomenon is known as bipolar or thermal seesaw[14,15]. It is also suggested that during Greenland's cooling and warming transitions, the global atmospheric teleconnections are characterized by a signal propagation from the northern high-latitudes to the tropics, and further, to the southern high-latitudes, which were synchronous within sub-centennial uncertainty[16–19]. Moreover, observational data points to an active role of hydroclimatic variations in the tropics and Southern Hemisphere around AHP and SAHP and other millennial-scale events[17,20–22]. Recent research has highlighted the important role of the decreased Amazon River runoff and associated sea-surface salinity anomalies in resuming AMOC[6], which in turn initiated the termination of AHP4. The noteworthy point is that the termination of SAHP4 [-38.62 ky BP (thousand years before present, where the present is 1950 CE)] predated the termination of AHP4 (-38.34 ky BP) and the associated Greenland warming transition (-38.34 ky BP) by hundreds of years[6]. Previous studies[6,17,20,22] point to a south-to-north signal propagation around the terminations of AHP and SAHP and other millennial-scale events suggesting a slow ocean response. Furthermore, during the early Last Glacial Maximum (LGM) (26 to 22 ky BP), the Greenland ice-core records have large age uncertainty ($\pm500-\pm800$ years)[2] and high-resolution absolutely-dated proxy records from low latitudes are still lacking[18]. These shortcomings largely impede the precise analyses of the climate signal propagation phase (north to south or vice-versa) inter-hemispherically. Consequently, it is yet unresolved whether the observed phasing remained consistent during the LGM[12,16,18]. Notably, LGM was an important period since it had a distinct boundary condition when global sea level and greenhouse gas ($CO_2$ and $CH_4$) concentrations were at their minima of the last glacial period (Supplementary Fig. 1)[23–25], which may have influenced the hemispheric coupling. AHP2, SAHP2 and HS2 were typical millennial-scale events during the LGM (Supplementary Fig. 1)[3,26], and it remains an important task to rigorously test the climate signal propagation and phase relationship between the events. Moreover, it remains poorly known how the global atmospheric circulation was tele-connected during HSs onsets, when there is a little-to-no imprint in Greenland ice-core $\delta^{18}O$. Therefore, an integrated understanding of hemispheric climate coupling including both high and low latitudes is still lacking.

In this study, we report nine speleothem $\delta^{18}O$ records from ASM and SASM domains spanning 27 to 22 ky BP, including HS2[3,26]. We present speleothem $\delta^{18}O$ records from the Indian summer monsoon (ISM) domain at a resolution of ~4 years. The age-model precision of our ISM $\delta^{18}O$ records is <50 years and each record was obtained by combining annual lamina counting and $^{230}Th$ dating results (see Methods). The chronological precision and resolution of our ISM records facilitate the quantification of the chronological biases in Greenland ice-core records via comparison with the Greenland dust flux record (as reflected by ice-core [$Ca^{2+}$]). The robustness of the ISM speleothem-based Greenland chronology is further supported by extant volcanic evidence and radiocarbon dates. Speleothem $\delta^{18}O$ records from the SASM domain are characterized by sub-centennial age-model uncertainties (<80 years) at the key intervals. Overall, we use speleothem $\delta^{18}O$ and bipolar ice-core records to understand the causal link between the high- and low-latitude climate systems in both hemispheres and to establish an interpretive framework of the rapid climate shifts during HS2/AHP2/SAHP2, their characterization, and manifestation.

## Results and discussion
### Speleothem samples
Two speleothem $\delta^{18}O$ records from Cherrapunji Cave (Cherrapunji-2 and Cherrapunji-2017-1), northeast India, in the ISM domain were used

to construct a composite Cherrapunji record (Fig. 1a). Six speleothem $\delta^{18}O$ records were obtained from six caves: Mawmluh Cave (MWS-1)[27], northeast India in the ISM domain; Yongxing Cave (YX-51)[28] in the East Asian summer monsoon (EASM) domain of China; Dongqinghe Cave (DQH-17) in the transitional zone between ISM and EASM domains of China as well as Marota (MAG)[29], Paixão (PX-07)[29], and Botuverá (BTV-4C)[30] caves in the SASM domain of Brazil. More precise $^{230}Th$ dates were obtained for a published speleothem $\delta^{18}O$ record (NAR-C) from Cueva del Diamante Cave, in the SASM domain of Peru[8]. Cave locations and domains are shown in Fig. 2f, g, Supplementary Fig. 2, and Table 1. Although there were different views in early studies regarding speleothem $\delta^{18}O$ interpretation in the ASM domain (e.g., refs. 31–34), recent developments have shown a general consensus that speleothem $\delta^{18}O$ variations on millennial-to-orbital timescales reflect the large-scale monsoonal circulation/rainfall, which is closely linked to the overall monsoon intensity, instead of local rainfall amount[7,35–37] (Supplementary Note 1.3). In sum, at millennial-to-orbital timescales, speleothem $\delta^{18}O$ records from ASM and SASM regions are a first-order reflection of the intensity of large-scale monsoonal rainfall/circulation and the north-south shifts of the ITCZ[7,8,27–30,36].

Speleothem chronologies are based on 127 $^{230}Th$ dates (Supplementary Data 1). The age models of speleothem Cherrapunji-2 and Cherrapunji-2017-1 were obtained by annual lamina counting using confocal microscopy (Supplementary Fig. 3) in combination with $^{230}Th$ dates (Supplementary Fig. 4a, b, see Methods). The age models of speleothem YX-51, PX-07, MAG, BTV-4C, and NAR-C were obtained using the StalAge algorithm[38] (Supplementary Fig. 5, see Methods). Different age-modeling schemes yielded essentially identical age models (Supplementary Fig. 5). A total of ~2810 stable oxygen ($\delta^{18}O$) isotope data were obtained (Supplementary Data 2). The spatial resolution of the measurements varies between 0.05 and 1 mm (see Methods). The comparison between the $\delta^{18}O$ records from the same and different caves in the same climatic region (Supplementary Fig. 6) suggests that the speleothem $\delta^{18}O$ records broadly replicate, although there are minor differences in their absolute values (Supplementary Note 1.4).

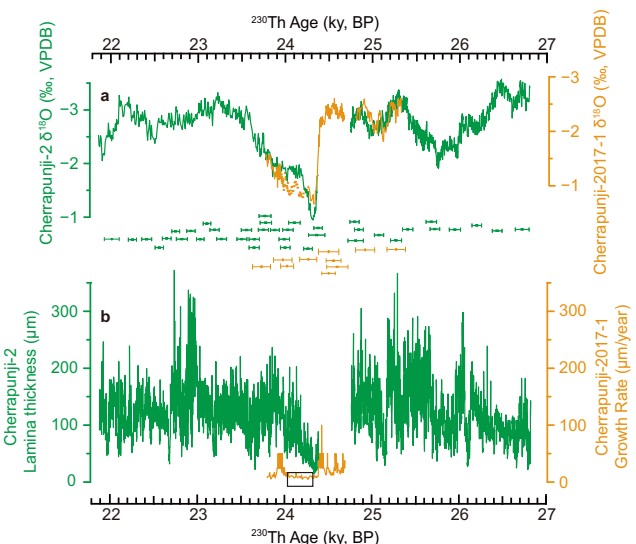

**Fig. 1 | Cherrapunji Cave speleothem $\delta^{18}O$ records. a** Speleothem $\delta^{18}O$ records from Cherrapunji Cave (Cherrapunji-2 and Cherrapunji-2017-1, this study) were used to construct a composite record. Over their contemporary growth intervals, we exclusively use the Cherrapunji-2 record (Supplementary Note 1.1). Error bars show $^{230}Th$ dates with uncertainties (2σ) for each record (color coded). **b** Annual lamina thickness of speleothem Cherrapunji-2 and estimated annual growth rate of speleothem Cherrapunji-2017-1. The black box in (**b**) shows the period with an extremely slow growth rate in Cherrapunji-2017-1, and the corresponding $\delta^{18}O$ data in (**a**) is depicted by dots.

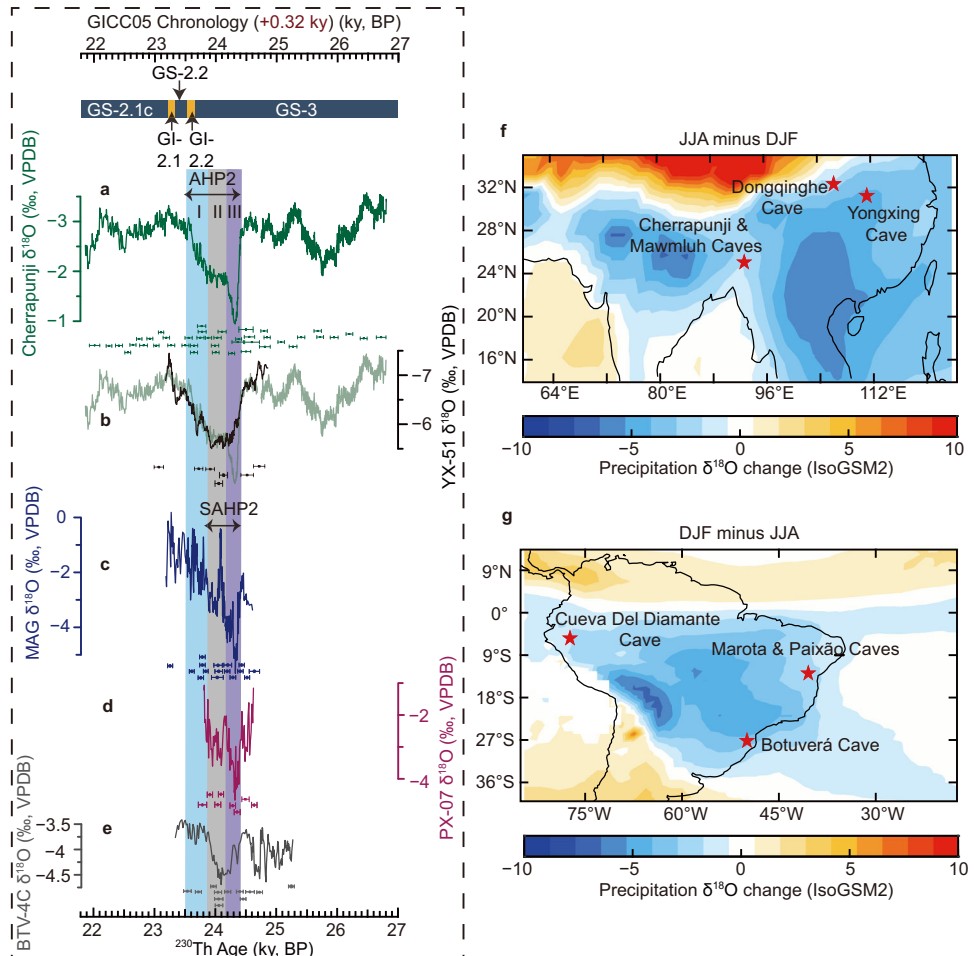

**Fig. 2 | Speleothem δ18O records from Asian summer monsoon and South American summer monsoon domains and cave locations. a–e** Speleothem δ18O records from Cherrapunji Cave (composite Cherrapunji record, this study) (**a**), Yongxing Cave (YX-51, black, this study) in comparison with the Cherrapunji record (light green) (**b**), Marota Cave (MAG, this study) (**c**), Paixão Cave (PX-07, this study) (**d**), and Botuverá Cave (BTV-4C, this study) (**e**). Note the inverted y-axis for (**a**) and (**b**). Error bars show 230Th dates with uncertainties (2σ) for each record (color coded). Greenland Stadial (GS) and Greenland Interstadial (GI)[2] are shown at the top. It is noted that the timing of Greenland events are based on the GICC05 chronology (Supplementary Note 1.5), which is shifted by +320 years (see main text). AHP2 and SAHP2 durations are marked by double-sided arrows. The vertical bars from left to right during AHP2 depict stages I, II, and III. **f** Spatial pattern of June-July-August (JJA) minus December-January-February (DJF) precipitation amount-weighted δ18O (‰) in the Asian summer monsoon domain using the Isotope-incorporated Global Spectral Model version 2 (IsoGSM2[93]) for 1979–2017. **g** is the same as in (**f**) but for DJF minus JJA in the South American summer monsoon domain.

## Table 1 | Cave locations and climate domains

| Region | | | Cave name | Location (latitude, longitude) | Speleothem name |
|---|---|---|---|---|---|
| Asian summer monsoon (ASM) domain | Indian summer monsoon (ISM) domain | | Cherrapunji | 25°11'59"N, 92°27'11"E | Cherrapunji-2 |
| | | | | | Cherrapunji-2017-1 |
| | | | Mawmluh | 25°15'44"N, 91°52'54"E | MWS-1 |
| | Transitional Zone between ISM and EASM domains | | Dongqinghe | 32°34'N, 106°12'E | DQH-17 |
| | East Asian summer monsoon (EASM) domain | | Yongxing | 31°35'N, 111°14'E | YX-51 |
| South American summer monsoon (SASM) domain | | | Marota | 12°35'S, 41°02'W | MAG |
| | | | Paixão | 12°37'S, 41°01'W | PX-07 |
| | | | Botuverá | 27°13'S, 49°09'W | BTV-4C |
| | | | Cueva del Diamante | 5°44'S, 77°30'W | NAR-C |

## Speleothem records in the ASM domain

The ASM is a vast climate system, composed mainly of the ISM and EASM subsystems, and closely coupled with the Asian westerly system[7,39–41]. The timing of the AHP2 hydroclimatic transitions in the ASM domain is tightly constrained by the Cherrapunji δ18O record at

high temporal precision (uncertainty <50 years) and resolution (~4 years) as well as its clear structure across AHP2 (Fig. 2a). Based on the Cherrapunji δ18O record, the onset of AHP2 is characterized by an abrupt increase in δ18O (-2‰) that began at 24.42 ± 0.04 ky BP (age model uncertainty) (2σ uncertainty bounds are used throughout this

work), and lasted for 76 ± 5 years (lamina counting uncertainty) (Supplementary Fig. 3a) before reaching the maximum value of AHP2 (Fig. 2a). A subsequent abrupt rebound commenced at 24.33 ± 0.02 ky BP and lasted 139 ± 2 years with a ~1‰ $\delta^{18}O$ decrease (Supplementary Fig. 3b), which, together with the abrupt onset, manifests as a positive $\delta^{18}O$ excursion during AHP2 (referred to as stage III) (Fig. 2a). Following the rebound a relatively stable period (stage II) occurred between 24.19 ± 0.02 ky BP and 23.87 ± 0.02 ky BP (Fig. 2a). The termination (stage I) was characterized by a gradual $\delta^{18}O$ decrease of ~1.2‰, which was initiated at 23.87 ± 0.02 ky BP and lasted for ~350 years (Fig. 2a). In order to objectively select the speleothem records in the ASM domain, we used the Speleothem Isotopes Synthesis and Analysis (SISAL) database[42,43] to choose the speleothem $\delta^{18}O$ records that have either a temporal resolution better than 40 years or a sub-centennial age uncertainty, and some high-quality ASM speleothem $\delta^{18}O$ records that are not in the database were also used in this study (Supplementary Fig. 7). The comparison between the Cherrapunji $\delta^{18}O$ record and other ASM domain speleothem records shows that their overall patterns and transitional timings across AHP2 are coherent (Fig. 2b and Supplementary Fig. 7). Notably, however, the stage III excursion is only evident in the records in the ISM domain and the transitional zone between the ISM and EASM domains, but is absent in the EASM domain (Fig. 2b and Supplementary Fig. 7), as discussed in following sections.

In this study, significant changes in the isotopic profiles were calculated using the "Ramp-fitting"[44] and "BREAKFIT" methods[45] (see Methods; Supplementary Fig. 8 and Supplementary Table 1). However, these two methods do not consider the age model uncertainty, thus in our analyses, we quadratically combined the change point uncertainty and the age model uncertainty and obtained a combined uncertainty (see Methods; Supplementary Table 2). The timing of the significant changes in the isotopic file of our raw dataset is further supported by the "Trend-fitting" function analyses (see Methods; Supplementary Fig. 9). We also performed these calculations to identify key change points in bipolar ice-core records (Supplementary Figs. 9, 10 and Supplementary Table 2).

## Greenland ice-core [Ca²⁺] records

Previous studies have shown that the Taklimakan and Gobi deserts, hereafter referred to as the Asian dust source regions (Supplementary Fig. 11), were the main contributors of mineral dust transported to Greenland during 27–23 ky BP[46,47] (Supplementary Note 1.6.1). In modern times, the main dust emission season in the Asian dust source region is boreal spring[48]. Dry conditions and strong winds in the dust source areas favor the entrainment of dust into the upper atmosphere, which can be long-distance delivered to the downwind North Pacific and Greenland via westerly jet[48,49]. During boreal spring, the upper westerly jet is strong and its core axis is located south of the Tibetan Plateau, while its meridional range is the largest[50] (Supplementary Fig. 11a). This scenario is distinct from boreal summer, when the upper westerly jet weakens and its core axis migrates to the north of the Tibetan Plateau, allowing the low-level monsoonal moisture to reach inland areas[51] (Supplementary Fig. 11b). Paleoclimatic studies further showed that during stadials, the cooling in the Northern Hemisphere (NH) weakens the seasonal contrast, extending the dust season from spring to summer and suggesting that the jet axis maintained above the southern part of the Tibetan Plateau for a longer time[41,52,53], which enabled more frequent dust emission and transport to Greenland. Based on these studies, we suggest that the variability of the Greenland ice-core mineral dust-derived [Ca²⁺] (a dust proxy[54], Supplementary Fig. 12) at decadal-to-millennial timescales during 27–23 ky BP primarily reflects the latitudinal position and intensity of the NH westerly winds, as well as the hydroclimate conditions in Asian dust source regions, consistent with the previous interpretation[55]. Such an argument gains support from the similarities between Greenland [Ca²⁺] and an Asian westerly domain speleothem record from Turkey (So-1 record

from Sofular Cave) for AHP2 (Supplementary Fig. 12). During the AHP2 onset, the So-1 $\delta^{13}C$ record exhibits an abrupt positive transition, indicating a change to the colder/drier condition[56], in line with a southward shift of the Westerlies. In the following discussion, Greenland ice-core [Ca²⁺] is qualitatively used as a proxy for mid-latitude westerly winds in the NH and hydroclimate conditions in the Asian dust source regions, which is different from the Greenland ice-core $\delta^{18}O$ records that mainly reflect NH high-latitude changes in temperature and precipitation seasonality.

## Correlation between Greenland ice-core [Ca²⁺] and ISM $\delta^{18}O$ records

Greenland ice-core $\delta^{18}O$ records do not exhibit a distinct pattern during HS2 (Fig. 3a). One of several possibilities is that due to an AMOC slowdown, the cooling signal (lower $\delta^{18}O$) was obfuscated by the absence of winter precipitation in response to sea-ice expansion, as suggested by transient experiments for HS1[57]. The blurred signal precludes a direct correlation between Greenland ice-core $\delta^{18}O$ records and the more precisely-dated ASM speleothem $\delta^{18}O$ records. Alternatively, the Greenland ice-core [Ca²⁺] record, as a dust proxy dominated primarily by Asian westerly winds that are strengthened rather than muted by sea-ice expansion[58], exhibits a clear structure across Greenland Stadial 3 (Fig. 3b). Crucially, the westerly winds are closely coupled to the ASM circulation and tracked by speleothem $\delta^{18}O$[40,59] on seasonal-to-orbital timescales[7,40,41,50,51]. We rely on this dynamic link to correlate Greenland ice-core [Ca²⁺] to ASM speleothem $\delta^{18}O$ records, assuming no time offset from lagged system responses at the decadal timescale. Indeed, Lidar data analyses and model simulations indicate that the transportation of modern dust is a fast process[60]. Modern Greenland snow-pit sample analyses and observations of dust-storm activities in the Asian dust source regions have further shown that no time offset exists between dust emission in the source region and dust deposition in Greenland at the interannual timescale (Supplementary Note 1.6.2).

A resemblance exists between Greenland ice-core [Ca²⁺] and Cherrapunji $\delta^{18}O$ records, which allows tuning the ice-core [Ca²⁺] timeseries [on the widely used Greenland Ice Core Chronology 2005 (GICC05)] to the ²³⁰Th-dated Cherrapunji $\delta^{18}O$ record using prominent peaks and abrupt changes as tie points (Fig. 3 and Supplementary Table 3). These tie points have passed sensitivity tests (see Methods) and have been verified by the "Trend-fitting" method (Supplementary Fig. 9a–c), and are hence regarded as statistically robust. Furthermore, five of the tie points in the detrended z-score transformed Cherrapunji $\delta^{18}O$ profile occurred at intervals with low z-score values (<−1) (Supplementary Fig. 13c), which we interpret to result from a higher frequency of extreme pluvials. Another tie point is at the start of AHP2, from which the Cherrapunji $\delta^{18}O$ values increase abruptly to the high z-score values (>4) (Supplementary Fig. 13c), which we interpret to result from a higher frequency of extreme droughts. These lines of evidence show that the six tie points correlate with large changes in Asian hydroclimate. Based on these tie points, the result suggests a shift of the GICC05 timescale by +320 years (Fig. 3), which is well within its quoted age uncertainty (the maximum counting error, a measure of the potential bias around this time interval is 600–800 years[2], Supplementary Fig. 14) and consistent with previous studies that also suggested that the GICC05 chronology needs to be shifted by several hundred years towards older ages[61,62]. Crucially, the +320-year shift is based on six tie points, much more precise than previous studies, which are merely based on one tie point with a larger age uncertainty (Supplementary Fig. 14). Moreover, although other millennial-scale ASM events (e.g., HS4/AHP4[6] and Younger Dryas[17]) apparently correlate with the ice-core [Ca²⁺] records (Supplementary Fig. 1), the dynamical mechanism underlying the correlation was not well explored in the previous studies. More importantly, no annually laminated speleothem record covers the full duration of 27–23 ky BP as of now, making it hard to precisely constrain the relative chronology, i.e., event

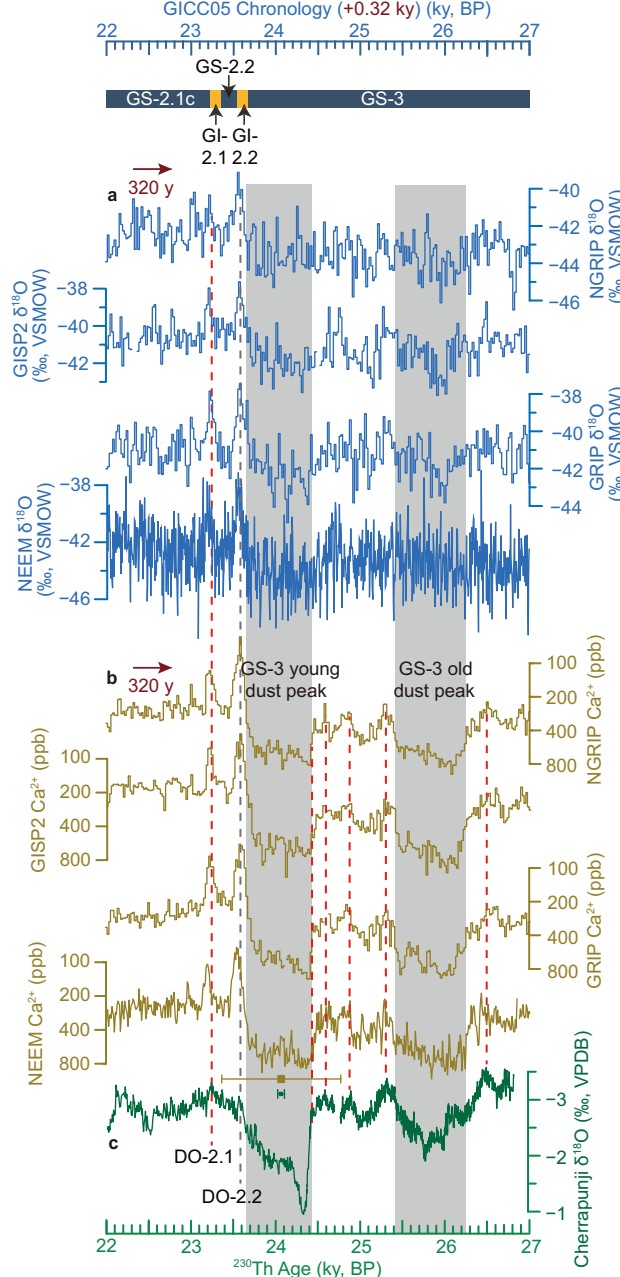

**Fig. 3 | Correlation between Greenland ice-core δ¹⁸O/[Ca²⁺] records and the Cherrapunji δ¹⁸O record. a** Greenland ice-core δ¹⁸O records: NGRIP, GISP2, GRIP, and NEEM records (Supplementary Note 1.5). **b** Greenland ice-core [Ca²⁺] records (note the inverted logarithmic [Ca²⁺] y-axis, as same as in ref. [2]): NGRIP, GISP2, GRIP, and NEEM records (Supplementary Note 1.5). **c** Cherrapunji speleothem δ¹⁸O record (this study). Greenland ice-core records are shown on the GICC05 chronology (Supplementary Note 1.5), which is shifted by +320 years via tuning the Greenland [Ca²⁺] time-series to the Cherrapunji δ¹⁸O record (see main text). Vertical red dashed lines depict the tie points between the ice-core and speleothem records (Supplementary Table 3). The leftmost two dashed lines correspond to Greenland Interstadial (GI)-2.1 and GI-2.2 peaks in Greenland[2] and their counterparts in the Asian summer monsoon regime (Dansgaard/Oeschger (DO)-2.1 and DO-2.2 peaks). Vertical gray bars depict the Greenland Stadial (GS)-3 dust peaks[94]. Error bars show uncertainty (2σ) of the ²³⁰Th chronology of the Cherrapunji record (green) in comparison to the possible age bias of the GICC05 timescale[2].

durations. In this regard, considering that the GICC05 timescale was confirmed to be precise within 20–40 years of uncertainty during Greenland Stadial 1 (~12.9–11.7 ky BP)[17], it is thus highly possible that ice layers were undercounted during Greenland Stadial 2 (~23–15 ky BP).

Applying the +320-year shift, the Greenland ice-core chronology has an uncertainty of 90 years (2σ), which comes from the age model errors of the Cherrapunji record plus a few other possible factors (Supplementary Note 1.7). We consider this uncertainty as a conservative estimate. Although ice-core [Ca²⁺] and speleothem δ¹⁸O records show an impressive match with each other, differences exist. For example, the Cherrapunji δ¹⁸O record exhibits more gradual transitions compared with ice-core [Ca²⁺] records, and the "double-spiked" Dansgaard-Oeschger (DO)-2 structure in the Cherrapunji δ¹⁸O record is relatively small (Fig. 3 and Supplementary Note 1.6.3). Clarifying these differences warrant further studies. Nevertheless, we argue that the abrupt shift of the large-scale atmospheric circulation (as reflected by the six tie points) was synchronous within multi-decadal uncertainty, especially considering that the ASM and the Asian westerlies are typically tightly-coupled[7,40,41,50,51]. This is also consistent with our proposition that the GICC05 timescale needs to be shifted as a whole between 27–23 ky BP without being stretched or compressed, which takes advantage of the annual layer-counted chronology of GICC05 and Cherrapunji record, ensuring the robustness of their direct comparisons.

### Correlations to Antarctic ice-core records

The highest-resolution CH₄[24] and ice δ¹⁸O records[13] available around Antarctic Isotope Maximum 2 were derived from the West Antarctic Ice Sheet (WAIS) Divide Ice-core (WDC) on a recent chronology (WD2014[63]). Remarkably, a unique volcanic triplet spike was identified in both Greenland and Antarctic ice-cores[12], possibly associate with the Oruanui eruption from the Taupo volcano (Supplementary Fig. 2)[12]. After shifting the GICC05 chronology by +320 years, the volcanic triplet spike occurred at 24,939 ± 90 y BP in Greenland (Fig. 4), providing a robust tie point that requires the WD2014 ice chronology (which shows the volcanic triplet spike at ~24,539 y BP) to be shifted by +400 years (320 years according to the GICC05 shift in this study and 80 years based on a shift suggested previously[12]) (Fig. 4). Moreover, after the +400-year shift, the tephra layer of the Oruanui eruption in WDC would occur at 25,718 y BP[64], consistent within uncertainty with the calibrated radiocarbon-age of 25,675 ± 90 (1σ) calendar y BP (using the SHCal20 calibration curve[65])[66]. Furthermore, the WD2014 gas chronology is also shifted by +400 years, particularly considering the small uncertainty of the WDC ice-gas age difference (Δage) at this time (~110 years)[63,67]. Ultimately, the adjusted chronology is linked to our speleothem chronology, and thus, should have a similar precision of ~90 years (2σ, ice chronology) provided that the volcanic triplet is correctly aligned. WD2014 is an annual layer-counted chronology between 31.2–0 ky BP[63], which does not allow being stretched or compressed beyond the expected error, consistent with the shift of WD2014 as a whole between 26–23 ky BP. Similarly, the WDC ice layers were possibly undercounted during the late LGM. As a result, this tuning allows a precise comparison between bipolar ice-core records and mid-to-low-latitude speleothem records on a common chronology. We acknowledge the potential existence of other matching strategies and the risk of circular reasoning. However, because abrupt transitions of the large-scale atmospheric circulations as well as speleothem and radiocarbon chronologies are aligned so well and are connected (Figs. 3, 4), our tuning process must have high reliability within our stated uncertainty (~90 years) and constructs the hitherto strongest constraint on bipolar ice-core chronologies between 27–23 ky BP.

Although bipolar ice-core chronologies have been improved, the analyses of the phase relationship between Greenland Interstadial 2 warming transition and Antarctic Isotope Maximum 2 cooling transition cannot yet be resolved (Fig. 4). This is because the small signal-to-noise ratio in the Antarctic ice-core δ¹⁸O record[13] impedes a robust identification of change points (see Methods). Nevertheless, the precise chronologies provide important constraints to resolve the problem as soon as Antarctic ice-core δ¹⁸O records with higher resolution are available.

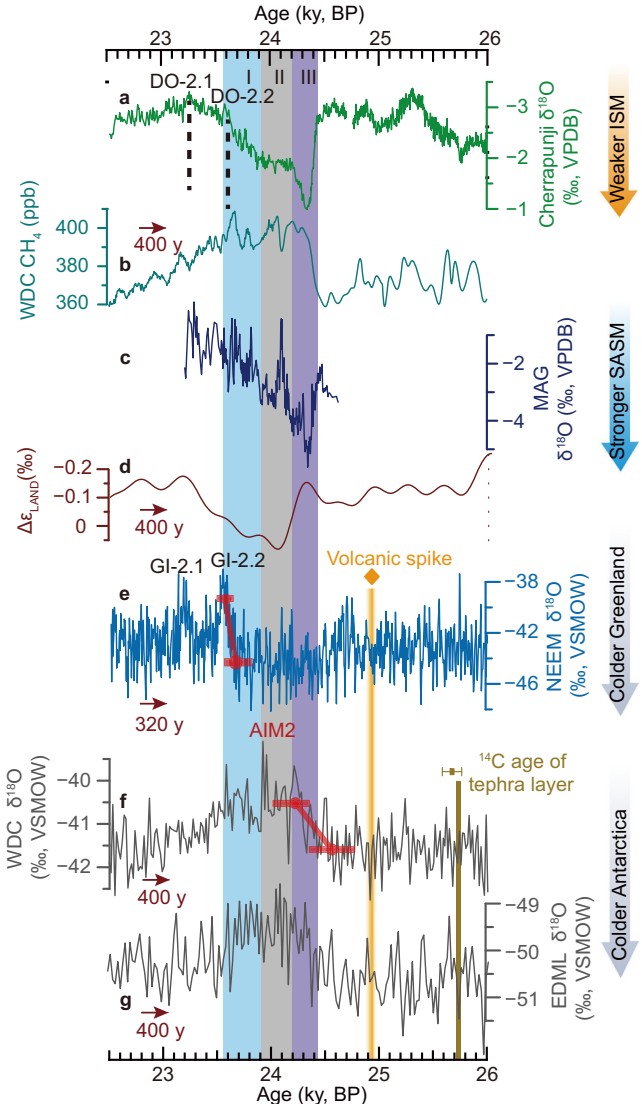

**Fig. 4 | Comparison between Asian summer monsoon/South American summer monsoon and ice-core records. a** Cherrapunji δ¹⁸O record (this study) from Cherrapunji Cave in the ISM domain, **b** Antarctica WDC ice-core CH₄ record[24], **c** MAG δ¹⁸O record (this study) from Marota Cave in the SASM domain, **d** Antarctic ice-core composite $\Delta\varepsilon_{LAND}$ proxy[71], **e** Greenland NEEM ice-core δ¹⁸O record (Supplementary Note 1.5) on the improved chronology (GICC05 timescale +320 years) (see main text), **f**, **g** are Antarctic ice-core δ¹⁸O records from WDC[13] and EDML[95], respectively. Antarctic ice-core records are plotted on the WD2014 chronology[63], which is shifted by +400 years via a correlation based on the volcanic triplet spike (depicted by the orange diamond and the vertical bar at -24.94 ky BP). It is noted that the EDML δ¹⁸O record has been tuned to the WD2014 chronology in a previous study[16]. Red dots and associated error bars in (**e**) and (**f**) indicate the timing and combined uncertainties (2σ) of the change points in Greenland and Antarctic ice-core δ¹⁸O records (Supplementary Table 2). The Brown error bar depicts the ¹⁴C age[66] and uncertainty (1σ) of the Oruanui eruption from the Taupo volcano. The vertical bar at -25.72 ky BP marks the tephra layer of the Oruanui eruption recorded in the WDC ice core[64]. The two vertical dashed lines depict the DO-2.1 and DO-2.2 peaks. Cave, ice-core and volcano locations are shown in Supplementary Fig. 2. Vertical bars are the same as in Fig. 2. AIM Antarctic isotope maximum[95], ppb parts per billion, ISM Indian summer monsoon, SASM South American summer monsoon.

## AHP2/SAHP2 onsets and the large excursion

A recent study[68] has established age models for the 92 published marine sediment records from the Atlantic Ocean via correlating to the Greenland ice-core GICC05 chronology. This, in turn, allows us to

simply shift the chronologies of these marine records by +320 years across the HS2 period (Supplementary Fig. 15 and Supplementary Note 1.8) to compare the marine records with the precisely-dated speleothem records.

It has been known that global atmospheric shifts are synchronous within decades at Greenland cooling and warming transitions, with a signal propagation from north to south[16–19]. However, only limited data are available about the global atmospheric teleconnections at HSs onsets when there are relatively small changes in Greenland δ¹⁸O records (e.g., ref. 69). In our temporal framework, the AHP2 onset at 24.43 ± 0.05 ky BP (combined uncertainty) inferred from the Cherrapunji δ¹⁸O record is synchronous within sub-centennial uncertainties with changes in the SASM hydroclimate (Fig. 5j and Supplementary Table 2). The abrupt and anti-phase hydroclimatic changes in the monsoon domains of the two hemispheres reinforce the hypothesis (e.g., ref. 70) that iceberg discharge or freshwater forcing originating from the NH subtropics could push the tropical rain-belt far southward. A positive anomaly can also be observed in the $\Delta\varepsilon_{LAND}$ record (Fig. 4d), which also supports a southward shift of the tropical rainfall and terrestrial oxygen production during the HS2[71,72]. This is also consistent with the observation that the weakening of the AMOC coincides with the waning of the ASM within the marine age uncertainties (Supplementary Fig. 15), similar to the results of water-hosing experiments[73]. Additionally, the large and abrupt transition provides a direct example of a fast and widespread monsoon climate swing that occurred on a decadal timescale, as constrained by lamina counting (i.e., onset in ~75 years, Supplementary Fig. 3a). Furthermore, we compared our records with proxy records from the Southern Hemisphere representing mid- to high-latitude atmospheric circulations and/or different moisture origins[16,19,74] (Supplementary Note 1.9), i.e., Antarctic ice-core $d_{ln}$ (the logarithmic definition of deuterium excess) and non-sea-salt soluble $Ca^{2+}$ ([nssCa²⁺]) records (Fig. 5h, i). The results show synchronous changes within uncertainties (Fig. 5j and Supplementary Table 2), suggesting parallel changes in Southern Hemisphere high- to low-latitude atmospheric circulations and NH mid-latitude atmospheric circulations (Fig. 5b–i). This synchronicity hints towards a fast (decadal-scale) atmospheric teleconnection, involving meridional shifts of the ITCZ and the mid-latitude westerly wind belts in both hemispheres, changes in the tropical Hadley circulation as well as monsoon circulations, which is dynamically consistent with the inter-hemispheric atmospheric teleconnections in model simulations[75,76].

After the abrupt AHP2/SAHP2 onsets, a rapid rebound is observed in Cherrapunji, PA-LA-1, MAG and PX-07 δ¹⁸O records (Fig. 5c–f). Notably, three of the four records are derived from tropical regions (Fig. 2f, g), and Cherrapunji Cave is close to the tropical regimes (Fig. 2f) with a major moisture trajectory from tropical oceans during summer monsoon months[36]. Further comparison shows that the rebound feature is absent in Southern Hemisphere mid- to high-latitude atmospheric circulation records (Fig. 5h, i), and the Southern Hemisphere subtropical record from Botuverá Cave (BTV-4C record) also does not exhibit a clear rebound structure (Fig. 5g). In the NH, Greenland [Ca²⁺] records do not exhibit a rebound feature (Fig. 3b), and the same is true for speleothem δ¹⁸O records from the EASM domain (Fig. 5b and Supplementary Fig. 7a–e). Additionally, the NGRIP deuterium-excess record, a proxy for the past changes in evaporation conditions or shifts in moisture sources in the NH mid-to high-latitudes[69], does not show a rebound feature (Fig. 5a). We, therefore, identify simultaneous changes in low-latitude regions without any fingerprint in mid-to high-latitude climates. The direction and speed of the rebound are nearly opposite to the onset transition, corresponding to a breakpoint during stage III (Fig. 5c–f). Change point detection results show that the breakpoints in our records (Cherrapunji, MAG and PX-07) are coherent within sub-centennial uncertainties (Fig. 5c, e, f and Supplementary Table 2), hinting at rapid atmospheric teleconnection within the tropics. The δ¹⁸O reversal in our monsoonal

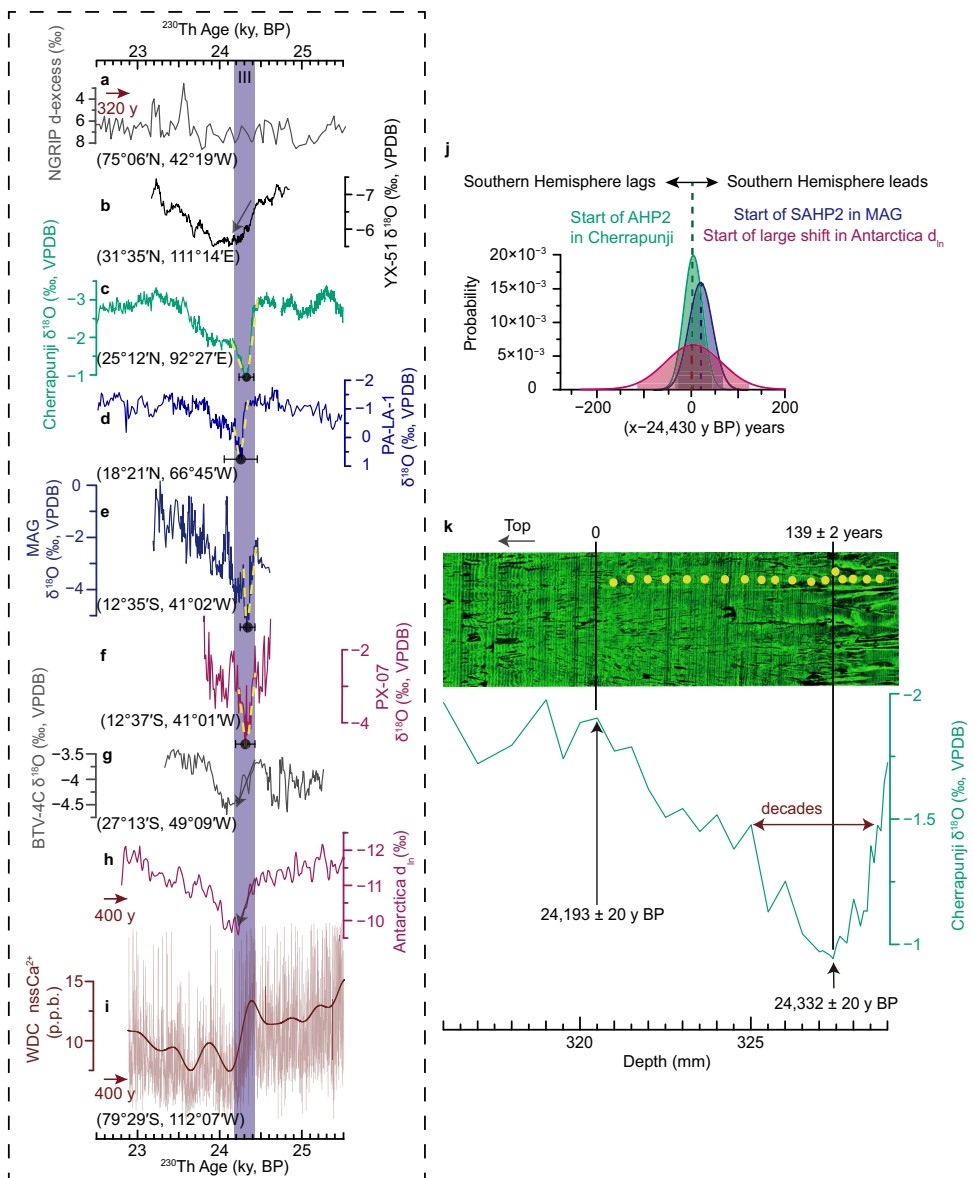

**Fig. 5 | Global atmospheric teleconnections during stage III. a** NGRIP deuterium-excess record[69] on the GICC05 chronology and shifted by +320 years.
**b–g** speleothem $\delta^{18}O$ records from Yongxing Cave (YX-51, this study), Cherrapunji Cave (Cherrapunji, this study), Larga Cave (PA-LA-1[96]), Marota Cave (MAG, this study), Paixão Cave (PX-07, this study), and Botuverá Cave (BTV-4C, this study), respectively. **h** Antarctica 5-core averaged $d_{ln}$ anomaly[16]. **i** Antarctica WDC ice-core non-sea-salt $Ca^{2+}$ [$nssCa^{2+}$] record[74] and overlain by the thick line (high-frequency variability (>1 cycle per 300 year) removed by a low-pass Butterworth filter, same as in ref. 74). Antarctic ice-core records in (**h**) and (**i**) are on the WD2014 chronology and shifted by +400 years. The yellow dashed lines in (**c–f**) are generated via the BREAKFIT algorithm[45]. Black dots and associated error bars in (**c–f**) depict the timing of breakpoints and combined uncertainties (2σ) (Supplementary Table 2). Latitudes and longitudes for the various records in (**a–g**) and (**i**) are shown. Note the inverted y-axis in (**a–d**) and (**h**). **j** Probability density functions of spatial age offsets between the start of AHP2 and SAHP2 as well as the start of the large shift in Antarctic ice-core $d_{ln}$ record (Southern Hemisphere records minus Asian summer monsoon record); shadings indicate the 95.4% intervals (2σ uncertainty), the uncertainties are the combined uncertainties (Supplementary Table 2). **k** Laser scanning confocal microscopy image of speleothem Cherrapunji−2 between 316–329 mm, the green curve is the plot of Cherrapunji-2 $\delta^{18}O$ versus depth; the timing of change points (bottom) and lamina counting results (top) are shown, the laminae were counted as ten between adjacent yellow dots, and the horizontal double-sided arrow depicts the rapid reversal which occurred within decades; the uncertainties of the change points in (**k**) depict the age model uncertainty (2σ) (Supplementary Data 2). AHP2 Asian Heinrich Period 2, SAHP2 South American Heinrich Period 2.

record is rapid, which occurred within decades as constrained by annual lamina data (Fig. 5k), manifesting the existence of a tipping point in the tropical climate system. We refer to this oscillation as a "tropical atmospheric seesaw". Oceanic processes would last longer than a few decades[17]. In this regard, the tropical atmospheric seesaw involves synchronous and opposite changes (positive $\delta^{18}O$ excursion in Cherrapunji and PA-LA-1; negative $\delta^{18}O$ excursion in MAG and PX-07) in monsoon domains of both hemispheres (Fig. 5c–f), possibly associate with the meridional shift of the ITCZ. It is noteworthy that

oceanic changes would accompany atmospheric changes in a coupled climate system, and our results may help to differentiate the roles of oceanic processes and atmospheric processes mentioned in this study, which is vital for understanding climate dynamics[13].

Although Greenland ice-core [$Ca^{2+}$] and EASM speleothem $\delta^{18}O$ records show a general coherent pattern with Cherrapunji $\delta^{18}O$ record across much of 27–23 ky BP, the NH mid-latitude changes were decoupled with ISM domain during stage III (Figs. 2b, 3b and Supplementary Fig. 7a–e). The mid-latitude records lack the prominent

excursion, which is exhibited when comparing the YX-51 and Cherrapunji records (Fig. 2b) (Supplementary Note 1.11). After stage III, the NH mid-to-low-latitude atmospheric circulations coupled again (Figs. 2b, 3b). This suggests that during AHP2 and SAHP2 onsets, the tropical atmospheric system amplified the disturbances from the north, and subsequently triggered the rebound after reaching a tipping point in the tropical hydroclimate. Moreover, the prominent excursion occurred during the warming phase of Antarctic Isotope Maximum 2 (Fig. 4f, g), while Greenland temperatures maintained low (Fig. 4e). Meantime, changes in low-latitude regimes suggest large-scale reorganizations of the atmospheric circulation, probably associate with meridional ITCZ shifts. In this context, the conventional bipolar seesaw concept[15] needs to be expanded in order to fully reflect the complexity of processes at play, including both high and low latitudes and various boundary conditions.

A previous study using a heuristic model of tropical rainfall distribution hypothesized that the $CH_4$ overshoots within HS1, HS2, HS4, and HS5 correlate with the onset of Heinrich events[24]. Although subsequent studies confirmed the correlation during some HSs by speleothem records of high-resolution and high-precision[6], the phase relationship between the ~40 ppb (parts per billion) $CH_4$ increase and Heinrich event 2 remains unclear (Supplementary Fig. 16). It is noteworthy that the previous relationship was derived by comparing the small WDC $CH_4$ peak (~15 ppb) with the GSIP2 $\delta^{15}N$ record (both in the gas phase, Supplementary Fig. 16), assuming that the $\delta^{15}N$ peak represents the DO-2 peak[24]. The ice-core $\delta^{18}O$ record, however, shows two abrupt warmings and associated peaks[2] (Fig. 4e), unlike the single DO-2 peak as observed in previous low-resolution data. Besides, the GISP2 $\delta^{15}N$ record bears very low resolution[2] (Supplementary Fig. 16). Therefore, the result would be different when matching the small WDC $CH_4$ peak with the DO-2.1 or DO-2.2 peak (Supplementary Fig. 16). Indeed, if based on the $CH_4$ record on the WD2014 chronology (i.e., without the +400-year shift proposed in this study), the abrupt $CH_4$ increase would correlate with the termination of AHP2 rather than its onset (Supplementary Fig. 16c, d), which is distinct from the previous assumption[24]. In our correlation framework, we confirm that the small $CH_4$ peak correlates with the DO-2.2 peak (Fig. 4a, b and Supplementary Fig. 16h, i). Moreover, the abrupt $CH_4$ increase (~40 ppb) coincides with stage III, the abrupt $\delta^{18}O$ shift shared by our records from the ASM and SASM domains (Fig. 4a–c and Supplementary Fig. 16h–j). This common signal provides compelling evidence that enhanced rainfall in the Southern Hemisphere tropical wetlands indeed coincided with a significant $CH_4$ increase early in AHP2, favoring the previous hypothesis[24].

## AHP2/SAHP2 terminations

During AHP2/SAHP2 terminations, a notable trait is a variance in trends between ASM and SASM records across stages I and II. The SASM speleothem $\delta^{18}O$ records exhibit a long-term trend across stage II showing a $\delta^{18}O$ increase (drying) in South America (Fig. 6a–d). After the termination of SAHP2 (~23.85–24.05 ky BP), all SASM speleothem $\delta^{18}O$ records exhibit maximum aridity, comparable to or even drier than pre-SAHP2 conditions (Fig. 6a–d and Supplementary Fig. 17g–j). In contrast, the ASM was still weak during stages I and II compared to the pre-AHP2 conditions, and AHP2 termination occurred ~300 years after the SAHP2 termination (Fig. 6f, g and Supplementary Fig. 17). Other records from the SASM domain (Supplementary Fig. 18c–e), however, lack sufficient $^{230}Th$ dates control around the SAHP2 in order to evaluate the transitional timing at sub-centennial precision. Nevertheless, the long-term drying trend during the SAHP2 termination, as observed in previous records (Supplementary Fig. 18c–e), is overall consistent with our observation (Supplementary Fig. 18a, b, f, g). The long-term drying trend in the SASM domain was possibly associated with a cooling of the tropical western equatorial Atlantic[77]. Additionally, a few centennial-scale events are superimposed on the SASM long-term

drying trend. Of note is a large dry event (inferred by a positive $\delta^{18}O$ excursion) in stage II in three SASM speleothem $\delta^{18}O$ records [(MAG and PX-07, this study, Fig. 6a, b) and arguably LSF-3 (Supplementary Fig. 18c)], in contrast to a rather stable climate in the ASM realm (Fig. 6e). Indeed, MAG, PX-07, and LSF-3 records are from caves located close to the tropical South Atlantic (Supplementary Fig. 2), and thus might be sensitive to changes in the surface thermohaline circulations nearby. Nevertheless, this dry event does not affect our conclusion of the much earlier termination in the SASM domain compared to the ASM domain, as demonstrated by various change point detection methods and sensitivity tests (see Methods, Supplementary Figs. 8, 9).

The observed long-term drying in the vast SASM domain would have caused a decrease in Amazon River discharge into the Atlantic Ocean[6] (Supplementary Fig. 19b). Reduced freshwater input may have induced a positive sea-surface salinity anomaly in the Amazon Plume Region[78,79], ultimately advected to the deep-water formation areas in the North Atlantic[80] (Supplementary Fig. 19b) and contributing to the strengthening of the AMOC[81,82]. The sea-surface salinity reconstructed for the eastern subpolar North Atlantic (MD95-2002 Core) indeed exhibits an increasing trend during the AHP2 termination (Supplementary Figs. 2, 15e). A strengthened AMOC would induce positive feedback via transporting more saline water to the north, facilitating northward heat transport and a northward shift in the tropic rain-belts, intensified ASM[6,17,22], and Greenland warming[14], consistent with the proxy data for stage I (Fig. 4a, e). Therefore, the temporal phasing established here shows that the initiation for the long-term drying in the SASM domain occurred hundreds of years before the AHP2 termination (the start of stage I) (Fig. 6f and Supplementary Fig. 17), consistent with marine records (Supplementary Fig. 2) and shedding light on the termination dynamics. The lead of the SASM in the termination process is prominent, which is based on the assessment of the combined uncertainties of proxy records (Fig. 6f, g, Supplementary Fig. 17, and Supplementary Table 2). This indicates that long-term drying in the SASM domain and associated positive sea-surface salinity anomalies in the Amazon Plume Region may have acted as a precursor for the AHP2 termination, which reinforces the hypothesis[6] recently proposed for the AHP4 termination, albeit having a different boundary condition compared to AHP2 (Supplementary Fig. 1). On the other hand, in the current theoretical framework, the hydroclimatic events surrounding the earlier termination of SAHP2 may have been causally linked to the hydroclimatic swerve (or the end of the Antarctic Isotope Maximum 2 warming transition) observed in Antarctic ice-cores (particularly the EDML ice core from the Atlantic sector) (Fig. 4 and Supplementary Fig. 19) (e.g., refs. 6,83), but the cause of the earlier Antarctic change itself requires further investigation.

In conclusion, our study provides a causal relationship between ASM speleothem $\delta^{18}O$ records and Greenland dust records, improving the bipolar ice-core chronologies by nearly an order of magnitude. Our observations support the notion that rapid atmospheric changes in the tropics is not merely a response to forcing from the north, since the amplification effect in the tropical atmospheric system is vital for the distinct isotope excursion, which is absent in mid- to high-latitudes. In terms of AHP2/SAHP2, the signal of tropical changes could be transported to the high-latitudes of both hemispheres only when the oceanic mode is dominant, such as a resumption of the AMOC. The emerging picture is that both atmospheric and oceanic processes in the tropical regions and the Southern Hemisphere played active roles during millennial-scale event terminations, superimposed on the bipolar seesaw. In this regard, our findings extend the bipolar seesaw theory in both temporal (including LGM) and spatial domains (including mid-to-low latitudes), which is vital for an integrated understanding of hemispheric coupling during rapid climate changes. This is also important for model simulations seeking to capture the climate dynamics of abrupt climate changes like Heinrich events.

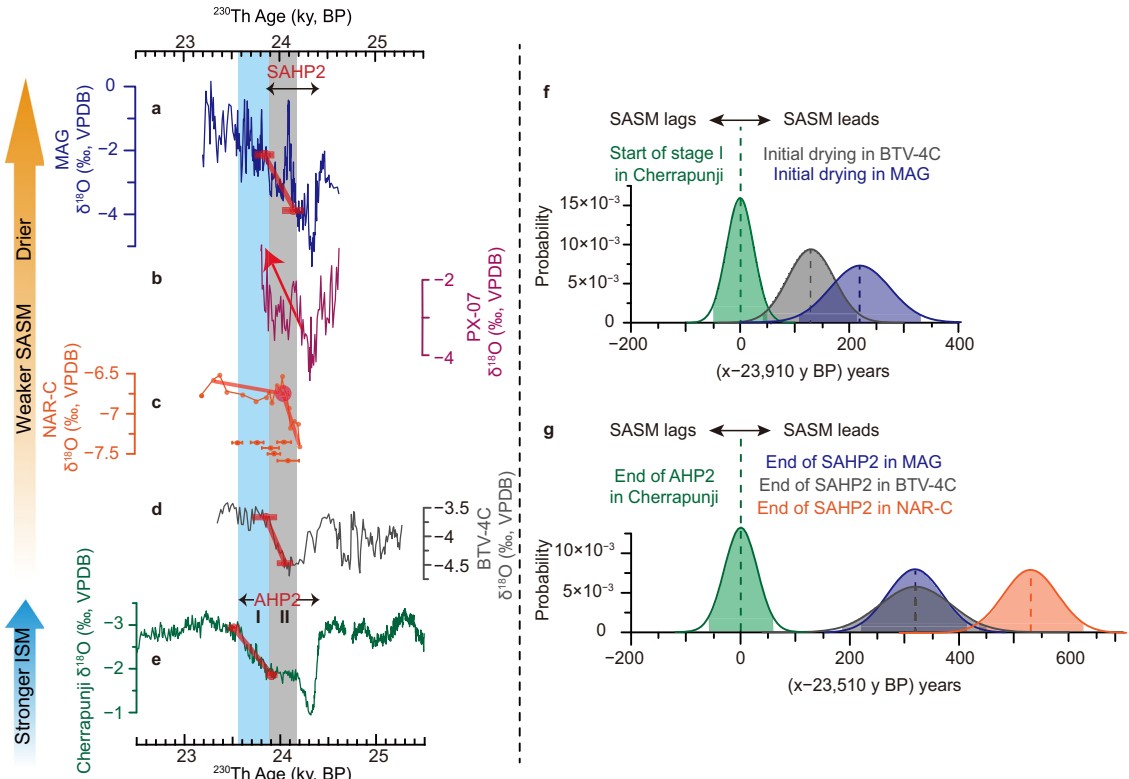

**Fig. 6 | Speleothem records from South America and their phasing relationship with Asian summer monsoon records during AHP2 and SAHP2 terminations.** **a–d** SASM speleothem δ¹⁸O records from Marota Cave (MAG, this study), Paixão Cave (PX-07, this study), Cueva del Diamante Cave (NAR-C, this study and ref. 8), and Botuverá Cave (BTV-4C, this study). **e** ASM speleothem δ¹⁸O record from Cherrapunji Cave (this study) (note the inverted y-axis). Cave locations are shown in Fig. 2f, g. Error bars in (**c**) depict ²³⁰Th dates and 2σ errors of the NAR-C record. Red dots and associated error bars in (**a**) and (**c–e**) indicate timing and combined uncertainties (2σ) of the change points in speleothem δ¹⁸O records (Supplementary Table 2), red lines in (**a**) and (**c–e**) show the ramps and break lines generated by

Ramp-fitting[44] and BREAKFIT[45] algorithms (see Methods). The red arrow depicts the trend in the PX-07 record during the SAHP2 termination. Vertical bars are the same as in Fig. 2. **f** Probability density functions of spatial age offsets between the start of stage I in ASM and the initial drying in SASM (SASM minus ASM) based on speleothem records reported in this study (Supplementary Table 2). Shadings indicate the 95.4% intervals (2σ uncertainty). **g** is the same as in (**f**) but for the comparisons at the end of AHP2 and SAHP2 (Supplementary Table 2). AHP2 Asian Heinrich Period 2, SAHP2 South American Heinrich Period 2, ASM Asian summer monsoon, SASM South American summer monsoon, ISM Indian summer monsoon.

## Methods

### Paleoclimate records

We reconstructed and considerably improved nine speleothem δ¹⁸O records. The sample information is summarized in Table 1. Cherrapunji Cave (-1100 m above sea level) and Mawmluh Cave are located at Cherrapunji on the southern fringe of the Meghalayan Plateau in Northeast India (Fig. 2f). The mean annual air temperature recorded at the nearby meteorological station in Cherrapunji is 17.4 °C. The mean annual rainfall in this region is -12,000 mm, 70% of which falls during the peak ISM months (June to September)[84] (Supplementary Fig. 7i). Dongqinghe Cave (-1600 m above sea level) is located at Guangyuan city, Sichuan province, Southwest China (Fig. 2f). The mean annual air temperature recorded at the nearby meteorological station is 16 °C. The mean annual rainfall in this region is -1100 mm, 70% of which falls during the summer monsoon months (June to September) (Supplementary Fig. 7i). Other caves are also located in the monsoon domains of both hemispheres[8,27–30].

### Proxy interpretation

The simulation work by ref. 36 examined the South Asian summer monsoon and showed that the interannual speleothem δ¹⁸O variability in India is strongly tied to the overall monsoon circulation. The intraseasonal to interannual rainfall variability over the Indian subcontinent exhibits a quasi-east-west precipitation dipole with anomalies of one sign over Northeast India and of an inverse sign over North, Northwest, and Central India[36,85]. The dynamical constraints[36]

explained the situation that the oxygen-isotope composition of precipitation (δ¹⁸O$_p$) of summer monsoon rainfall at Northeast India, albeit being located at the opposite end of the precipitation dipole, reflects upstream changes in monsoon precipitation amount over North, Northwest, and Central India (-15–28°N and 70–84°E) (via the moisture source effect instead of the "classical" amount effect). Following these reasonings and given the resemblance between Cherrapunji δ¹⁸O and MWS-1 δ¹⁸O (from Mawmluh Cave, which is located in the same climate region as Cherrapunji Cave) records (Fig. 2f and Supplementary Fig. 6d), we interpret low and high δ¹⁸O values in the Cherrapunji record to reflect strong and weak large-scale monsoonal circulation, respectively. At millennial timescale, the interpretation of other speleothem δ¹⁸O series utilized herein are consistent with the intensity of large-scale monsoonal rainfall/circulation and the north-south shifts of the ITCZ[7,8,27–30]. Supplementary Note 1.3 gives a detailed description of the ASM domain speleothem δ¹⁸O interpretation.

### ²³⁰Th dating method

Subsamples for ²³⁰Th dating were drilled on the polished speleothem section using a 0.3-mm carbide dental drill. We used standard chemistry procedures[86] to separate U and Th. A triple-spike (²²⁹Th-²³³U-²³⁶U) isotope dilution method was used to correct instrumental fractionation and to determine U-Th isotopic ratios and concentrations[87,88]. ²³⁰Th dating was performed at Xi'an Jiaotong University, China, using a Thermo-Finnigan Neptune Plus multi-collector inductively coupled plasma mass spectrometer (MC-ICP-MS). U and Th isotopes were

measured on a MasCom multiplier behind the retarding potential quadrupole in the peak-jumping mode using standard procedures[87]. Uncertainties in U and Th isotopic measurements were calculated offline at the 2σ level, including corrections for blanks, multiplier dark noise, abundance sensitivity, and contents of the same nuclides in the spike solution. The most recent values of the decay constants of $^{234}$U[88] and $^{230}$Th[88] and $^{238}$U[89] were used. Corrected $^{230}$Th ages assume an initial $^{230}$Th/$^{232}$Th atomic ratio of $(4.4 \pm 2.2) \times 10^{-6}$ and those are the values for material at secular equilibrium with the bulk earth $^{232}$Th/$^{238}$U value of 3.8[88]. The corrections for samples in this study are small because their uranium concentrations are high (~200–2500 ng/g) and detrital $^{232}$Th is low (less than 4000 pg/g) (Supplementary Data 1). Moreover, some studies applied an initial $^{230}$Th/$^{232}$Th ratio of $(8.8 \pm 8.8) \times 10^{-6}$ to provide a maximum uncertainty estimate for speleothem datasets all over the globe[18]. This value is too large for our clean samples, nevertheless, we recalculated our $^{230}$Th ages using the $^{230}$Th/$^{232}$Th ratio of $(8.8 \pm 8.8) \times 10^{-6}$ (Supplementary Data 1). The comparison shows that the recalculated ages are broadly similar to the original ages for our speleothems within uncertainty and do not show considerable changes (Supplementary Data 1). Therefore, in this study, a $^{230}$Th/$^{232}$Th ratio of $(4.4 \pm 2.2) \times 10^{-6}$ is applied to the final age models.

## Stable-isotope analyses

A total of ~2810 subsamples were drilled for stable-isotope analyses ($\delta^{18}$O and $\delta^{13}$C). About 313 subsamples were micro-milled at a spatial resolution of 0.05 mm from 40.5 to 55.4 mm along the central axis of speleothem Cherrapunji-2017-1, 331 were micro-milled at a spatial resolution of 0.1 mm from 19.7 to 59.2 mm for speleothem YX-51, 1321 from Cherrapunji-2 (spatial resolution varies between 0.1 and 1 mm), 74 from PX-07 (1 mm resolution), 199 from MWS-1 (0.2 mm resolution), 284 from MAG (1 mm resolution), 157 from BTV-4C (0.2 mm resolution) and 127 from DQH-17 (0.5 mm resolution). All subsamples were analyzed using a MAT253 isotope ratio mass spectrometer equipped with a Multi Prep system at Xi'an Jiaotong University, China. $\delta^{18}$O values are reported in per mil (‰) deviations, relative to the Vienna Pee Dee Belemnite (VPDB) standard, $\delta^{18}\text{O} = [(^{18}\text{O}/^{16}\text{O}_{sample})/(^{18}\text{O}/^{16}\text{O}_{VPDB}) - 1] \times 1000$. The analytical precision of the $\delta^{18}$O and $\delta^{13}$C analyses is better than 0.1‰ (1σ) (Supplementary Data 2).

## Annual lamina counting

In order to construct a precise floating lamina chronology (counting chronology), the annual nature of the laminae needs to be established. This can be achieved by comparing the number of laminae counted between absolutely-dated ages. After polishing, samples Cherrapunji-2 and Cherrapunji-2017-1 have clear lamina bands observed using confocal laser fluorescence microscopy (CLFM) (Supplementary Fig. 3). In this study, we utilized a Nikon A1-1024 instrument from the State Key Laboratory for Manufacturing Systems Engineering, Xi'an Jiaotong University. A 40 mW, 488 nm blue laser line was used for excitation[90], and fluorescence images were collected using an emission filter which allows light with wavelengths between 500 and 550 nm (visible, green) to pass through[90]. Lamina counting was conducted five times (Supplementary Note 1.2). The result shows that the laminae can be continuously identified, and the numbers of laminae between different depths are identical to the age difference of the respective $^{230}$Th dates within analytical uncertainty (Supplementary Data 1). This agreement supports our interpretation that the laminae are annual[91] and allows constructing a precise age model[92].

## Age models

Age models for annually laminated speleothems were reconstructed using the floating lamina chronology anchored by 39 and 9 high-precision $^{230}$Th dates for samples Cherrapunji-2 and Cherrapunji-2017-1, respectively (Supplementary Data 1 and Data 2) (Supplementary Fig. 4), using the least-squares fitting method to get the best fit of the

$^{230}$Th dates to the floating lamina chronology[92]. These age models have an estimated maximum age uncertainty of ±21 years (2σ, 99.5–329.5 mm in Cherrapunji-2), ±23 years (2σ, 331.5–437 mm in Cherrapunji-2), and ±43 years (2σ, 40.5–56 mm in Cherrapunji-2017-1) (Supplementary Data 2). The age model constructed using this method considered counting uncertainty and the uncertainty when anchoring the floating lamina chronology to the absolutely-dated $^{230}$Th dates[92]. This method does not depend on the analytical uncertainties of $^{230}$Th dates, thus the uncertainty of the age model is generally smaller than the dating uncertainty (Supplementary Data 2). We used StalAge algorithm[38] to construct the age models for YX-51, MAG, PX-07, BTV-4C and NAR-C (Supplementary Fig. 5). StalAge is particularly suited for speleothems creating objective age models based on two assumptions: (1) the age model is monotonic and (2) a straight line is fitted through all data or through as many data points as possible within error bars[38]. StalAge produces 300 realizations of age models by the Monte-Carlo simulation to account for the 95% confidence limits[38]. Major outliers are identified by disagreement with at least two data points, and minor outliers are screened if more than 80% of the simulated straight lines fail to have a positive slope. In our case, no major or minor outliers were detected because all ages in each age model increase monotonically within dating uncertainties (Supplementary Fig. 5). The uncertainty is large (more than 150 years) in the age model boundary of sample BTV-4C, YX-51 and NAR-C, due to limited dating controls. Nonetheless, the key change points (the onset and termination of AHP2/SAHP2) are robust, which is away from the age model boundaries and well constrained by several $^{230}$Th dates (Supplementary Fig. 5). Moreover, age models of our speleothem records were also calculated using other age-modeling schemes (Supplementary Fig. 5). All these schemes yielded nearly identical results and the conclusions of this study are thus insensitive to the choice of the age model (Supplementary Fig. 5).

## Change point determination

In order to objectively identify change points in various records, we used the Ramp-fitting[44] and BREAKFIT[45] algorithms. The choice of method is based on the shape of the time-series (Supplementary Table 1). "Ramp-fitting" finds two change points ($t_1$ and $t_2$), it estimates two constant levels of pretransition ($t > t_2$) and post-transition ($t < t_1$) and fits a ramp between them; Each data series was calculated for $3 \times 10^6$ steps using a Markov Chain Monte-Carlo (MCMC) sampler. These features make Ramp-fitting suitable for BTV-4C, Antarctica $d_{ln}$ and Greenland $\delta^{18}$O records, as well as the AHP2/SAHP2 termination process in Cherrapunji and MAG $\delta^{18}$O records. Both EDML and WDC $\delta^{18}$O records are characterized by an initial increase, a plateau, and subsequent decrease across the Antarctic Isotope Maximum 2 (Fig. 4), suitable for the Ramp-fitting routine in this case. Ramp-fitting also evaluated the influence of the addition of autocorrelated noise on the identified transitions, which in turn creates a more conservative estimate (with larger uncertainties) in the ramp parameters than other similar change point detection methods. BREAKFIT finds one change point and fits a linear slope on either side; these features make it suitable for the large excursion in Cherrapunji, MAG, PX-07 and PA-LA-1 $\delta^{18}$O records, as well as the determination of tie points between Cherrapunji $\delta^{18}$O and Greenland [Ca$^{2+}$] records. The BREAKFIT algorithm can provide statistical uncertainties of the timing of breakpoints using 2000 block bootstrap simulations, it also considers autocorrelation coefficients for the case of uneven time spacing[45]. In this study, we report the uncertainties of change points based on Ramp-fitting and BREAKFIT methods, and the analytical time intervals are shown in Supplementary Table 1. The main criteria to choose analytical time intervals are as follows: (1) the interval contains two prominent change points for Ramp-fitting and (2) the same time intervals are used for records from the same region if possible. To assess the sensitivity of the identified change points, we performed tests in which we

randomly change the search time windows (Supplementary Figs. 8, 10). The results are regarded as robust only in the case where these techniques provide an unequivocal solution, i.e., the timing of the identified change points do not vary by more than 60 years when the width of the search time window changed (Supplementary Fig. 8). The results show that most change points are insensitive to the choice of search time windows, one exception is the Antarctic Isotope Maximum 2 cooling transition in Antarctic ice-core $\delta^{18}O$ records where the identified change points do not meet the aforementioned criteria (Supplementary Fig. 10).

It is noteworthy that in most cases, our search time intervals meet the requirements of Ramp-fitting (contains two prominent change points) and BREAKFIT (contains one obvious change point), and no cropping is needed. However, the large dry event in the MAG record makes it hard to obtain the overall transitional trend during the SAHP2 termination and associated change points; in this scenario, we need to crop this event (Supplementary Fig. 8c). The cropped sections of the MAG raw data are replaced with constant values which equal the boundary values of the uncropped part of the record (Supplementary Fig. 8c).

In order to further check the robustness of the identified change points, we used another method by fitting "trends" to the whole time-series which does not require setting the search time window in advance (namely "Trend-fitting" method) (Supplementary Code 1). This method does not calculate the uncertainties for change points. The Trend-fitting results are shown in Supplementary Fig. 9. In this study, we only display results that have passed the sensitivity test and are confirmed by "Trend-fitting" method, i.e., the change points picked by "Trend-fitting" method fall within the uncertainties determined via BREAKFIT and "Ramp-fitting" algorithms. In our case, the change points derived via different methods are comparable (Supplementary Fig. 9). The exceptions are the Antarctic Isotope Maximum 2 cooling transition in the WDC $\delta^{18}O$ record and the Antarctic Isotope Maximum 2 transition in the EDML $\delta^{18}O$ record (Supplementary Fig. 9), which are excluded in our analyses.

### Combined uncertainty
In order to conduct a conservative estimate, we calculated the combined uncertainties for critical change points. Because Ramp-fitting and BREAKFIT merely calculate the change point uncertainty (Supplementary Table 2), we then quadratically combined the change point uncertainty and the age model uncertainty ($\sqrt{x^2 + y^2}$) to obtain the combined uncertainty (Supplementary Table 2). We contend that these uncertainties should be regarded as conservative estimates.

### Additional test
Considering the age models could be asymmetrical and may influence the change points, we performed an additional test to check this situation. In this regard, we conducted the BREAKFIT and Ramp-fitting analyses for our speleothem records using the 2.5$^{th}$ and 97.5$^{th}$ percentile of age model ensemble of each record (Supplementary Table 4). We then calculated the average of the change points derived via these two age models and compared it with the change point derived via the median (50$^{th}$) percentile of age models (Supplementary Table 4). The result shows that the asymmetry of age models does not significantly influence the change points of our speleothem records (Supplementary Table 4) as they are all constrained by several precise $^{230}$Th dates, the conclusion of this study is therefore unaffected by this problem.

### Data availability
The absolute $^{230}$Th dates for the 9 speleothems are provided in Supplementary Data 1, the $\delta^{18}O$ time-series for 9 speleothem records, the $\delta^{13}C$ time-series for speleothems from Cherrapunji Cave as well as the annual lamina thickness data of speleothem Cherrapunji-2 are provided in the Supplementary Data 2, other data from the referenced papers shown in the main figures are provided in the Supplementary Data 3. The data to support all the analyses in this study have been deposited in the NOAA repository (https://www.ncdc.noaa.gov/paleo/study/36854).

### Code availability
The "Trend-fitting" analyses were performed using MATLAB (R2019b). The MATLAB codes used in the "Trend-fitting" analyses are provided in Supplementary Code 1.

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

## Acknowledgements

This work was supported by the National Nature Science Foundation of China (NSFC grants 41731174, 41888101, and 42150710534 to H.C., NSFC grants 41703007 and 42172201 to G.K.), the Strategic Priority Research Program of the Chinese Academy of Sciences (Grant No. XDB 40000000) to Z.A., the FAPESP 2017/50085-3 to F.W.C., and the Carlsberg Foundation via the grant to SOR's ChronoClimate project. We thank Zhengyao Lu from Lund University, Sweden, for her great help. We thank Xiaoli Qu, Xianglei Li, Jiayu Lu, Youwei Li, Pengzhen Duan, Xue Jia, Baoyun Zong, Qiang Li, Yuan Yao, Jian Wang, Rui Zhang, Shouyi Huang, Binglin Meng, Xu Gao, and Lijuan Sha from Xi'an Jiaotong University for their persistent help. We thank Liangcheng Tan, Xing Cheng, and Peng Cheng from the Institute of Earth Environment, Chinese Academy of Sciences, for their discussions.

## Author contributions

G.K. and H.C. conducted the fieldwork. H.C. conceptualized this study. X.D., Y.X, Y.N., and F.Z. carried out the experiments and data analyses. N.M.S., S.C., X.W., A.K.G., S.D., and F.W.C. helped organize fieldwork and sampling. X.D., H.C., G.K., H.L., H.Z., Y.C., C.P.-M., and J.Z. interpreted results, H.C., X.D., G.K., S.O.R., A.S., J.P.S., A.S., Z.S., J.B., C.S., A.C., Z.A., and R.L.E. made revisions, H.C. and X.D. accomplished the writing with the help of all co-authors.

## Competing interests

The authors declare no competing interests.

## Additional information

[1]Institute of Global Environmental Change, Xi'an Jiaotong University, Xi'an 710049, China. [2]Physics of Ice, Climate and Earth, Niels Bohr Institute, University of Copenhagen, Copenhagen 2100, Denmark. [3]Scripps Institution of Oceanography, University of California San Diego, La Jolla, CA 92093, USA. [4]Department of Earth Science, California State University, Carson, CA 90747, USA. [5]State Key Laboratory of Loess and Quaternary Geology, Institute of Earth Environment, Chinese Academy of Sciences, Xi'an 710061, China. [6]Center for Excellence in Quaternary Science and Global Change, Chinese Academy of Sciences, Xi'an 710061, China. [7]Institute of Geology, University of Innsbruck, 6020 Innsbruck, Austria. [8]Department of Earth Sciences, University of Pisa, Via Santa Maria 53, 56126 Pisa (PI), Italy. [9]Department of Geochemistry, Universidade Federal Fluminense, Niterói 24020-141, Brazil. [10]School of Geography, Nanjing Normal University, Nanjing 210023, China. [11]Key Laboratory of Virtual Geographic Environment (Nanjing Normal University), Ministry of Education, Nanjing 210023, China. [12]Jiangsu Center for Collaborative Innovation in Geographical Information Resource Development and Application, Nanjing 210023, China. [13]Earth Observatory of Singapore and Asian School of the Environment, Nanyang Technological University, Singapore 639798, Singapore. [14]Department of Geology and Geophysics, Indian Institute of Technology Kharagpur, Kharagpur, India. [15]Wadia Institute of Himalayan Geology, Dehradun 248001, India. [16]Instituto de Geociências, Universidade de São Paulo, São Paulo 05508-090, Brazil. [17]Department of Earth and Environmental Sciences, University of Minnesota, Minneapolis, MN 55455, USA. [18]Key Laboratory of Karst Dynamics, MLR, Institute of Karst Geology, CAGS, Guilin 541004, China. ✉e-mail: kathayat@xjtu.edu.cn; cheng021@xjtu.edu.cn

