## [Peer Review File · Nature Communications]

Coupled atmosphere-ice-ocean dynamics during Heinrich Stadial 2REVIEWER COMMENTS

Reviewer #1 (Remarks to the Author):

General comments:

Dong et al. present a new study that investigates the changes in atmospheric and oceanic processes in the tropical regions and Southern Hemisphere during the HE2 millennial event. They perform new oxygen isotopic measurements on eight speleothem records from mid to low latitudes (Asia and South America) at multi-annual resolution over HE2 as well as ^{230}Th dating. Based on their speleothems records and other constraints, the authors proposed to revise the ice core chronologies of +320 y for Greenland and +400 y for Antarctica, which allows for a synchronous HE2 onset globally. They also investigate the monsoon dynamic during this period and proposed to use the Greenland Ca^{2+} concentration record as a qualitative proxy for change in mid-latitude westerly winds position and intensity in the Northern hemisphere, and hydroclimate conditions in the Asian dust source region.

The paper is well-written and complete, with many aspects addressed. The work is technically rigorous and the methodology is exhaustive, with the different sensitivity tests performed and details on the construction of the new chronologies. The supplementary information is however a bit dense for the reader with many figures. This study is interesting and worth to be published in Nature Communications. I do have small comments which I hope can help improve the manuscript.

The term “Heinrich event” is probably not appropriated for this study as it refers to the cooling of the sea surface temperature synchronous with the IRD inputs observed in the North Atlantic. Since the authors present records in the mid-low latitudes with no direct evidence of IRD deposition, I would suggest to use the term “Heinrich stadial” (in the main text and figures) that denotes the cold phase in which occurred the Heinrich event (Barker et al., 2009).

The authors interpret the speleothem $\delta^{18}\text{O}$ record as a proxy for monsoon circulation in India but the interpretation of this proxy is still in debate. Some studies interpreted the speleothem $\delta^{18}\text{O}$ as a proxy for monsoon intensity (related to precipitation changes in response to external climate forcing – e.g. Wang et al., 2008; Cheng et al., 2009), whereas others suggest that the speleothem $\delta^{18}\text{O}$ should not be considered as a summer monsoonal proxy only and that it reflects changes in moisture sourcing (e.g. Pausata et al., 2011; Caley et al., 2014). This should be clarified in the manuscript.

My main concern is about the comparison between speleothem $\delta^{18}\text{O}$ records and marine records in the North Atlantic Ocean that deserves to be more developed than the few sentences in the Supplementary text 1.9 about the adaptation of the sediment cores chronology with older age by 320 y. It would be nice to make the relationship between the changes in the low latitude and in the North Atlantic (climate, oceanic circulation...) and to discuss about the improvement of the chronology for these archives. The authors discussed in the main text about AMOC changes associated with changes in sea-surface salinity following changes in precipitation but don't refer to any marine records to support these hypothesis. Water-hosing modelling experiments also exist to show the response of the monsoon wind and precipitation in East Asia following a reduction of the AMOC (e.g. Sun et al., 2011). These two aspects worth to be developed in the discussion.

Other proxies of the low latitude hydrological cycle exist like the $\delta^{18}\text{O}_{\text{atm}}$ (at high resolution in Siple Dome ice core over the time period considered in this manuscript – Severinghaus et al., 2009). To support the active role of the low latitude hydroclimate dynamics during millennial events in the discussion, a comparison between the speleothem records and the $\delta^{18}\text{O}_{\text{atm}}$ could be added since positive anomalies are observed during Heinrich stadials, following a shift of the ITCZ.

Specific comments:

Line 102: precise “of the two monsoon systems”.

Line 105: instead of “muted” I would rather say “relatively small”.

Line 130: an introduction of the AHP2 and SAHP2 is needed.

Line 135: what about speleothem $\delta^{18}\text{O}$ records from South American monsoon domain?

Line 151: Fig. 1B and 1C.

Line 160: precise the resolution of the measurements.

I also wouldn't say that there is an “excellent replication between $\delta^{18}\text{O}$ records from the same cave” since there is a shift in the absolute $\delta^{18}\text{O}$ value between previous and new measurements for Yongxing and Paixão caves, even if they both show the same variations (see comment on Supplementary text 1.3).

Line 236: either the z-score value is >-1 or a tie point is missing on Supplementary Fig. 12 to show the tie point with a z-score value >4 .

Line 318: delete one “monsoon” term.

Lines 368-369: is it possible to state on the role of the ITCZ position on the CH_4 and to decipher the role of low latitude wetlands vs high latitude emissions?

Line 462: precise that the $\delta^{13}\text{C}$ measurements are not shown in this paper.

Lines 523-524: it is not clearly explained how the search time windows used for the Ramp-fitting technique in Supplementary Fig.7 and 9 are defined. Even if the text indicates that they are randomly chosen, I guess that there are still some early constraints.

Supplementary text 1.2: why the counting for Cherrapunji-2017-1 is manual and not machine-assisted?

Supplementary text 1.3: previous and new $\delta^{18}\text{O}$ measurements for Yongxing and Paixão caves have been done on the same speleothem. So when comparing the previous and new measurements I don't expect a large shift between the two, as observed in Supplementary Fig. 5a between 24.5–24 ka and in Supplementary Fig. 5c between 24.5–25 ka. This need to be explained (analysis not done in the same laboratory, with different protocole/instrument, fractionation effect...).

Supplementary text 1.5.2: “...during the ADS dust season...” please rephrase.

In the sentence “between dust emssion in ADS region”, correct the word “emission”.

Supplementary text 1.5.3: is there a reason for similar Greenland Ca^{2+} concentration with smaller amplitude in $\delta^{18}\text{O}$ during 26.5–25.3 ka than during AHP2? Could the Greenland record not only be linked to changes in the asian dust source (for example with another influence from the Northern Hemisphere)?

Supplementary text 1.6: I would also have considered an additional uncertainty₄ associated with the ^{230}Th dating method to obtain the Cherrapunji record chronology.

Supplementary text 1.7: explain that the d_{In} is the logarithmic definition of deuterium excess that better preserves isotopic moisture source information than the linear definition.

In the sentence "...it is commonly used as a proxy to trace...", remove "as a proxy".

Supplementary Fig.2: precise the range of "large uncertainties" of the chronology of marine records.

Supplementary Fig.10: correct the caption to "Monthly data are from...".

The ITCZ position could also be added for both months.

Supplementary Fig.11: the $\delta^{13}\text{C}$ has been measured in the speleothems of this study. Why not showing them here instead of the So-1 record that doesn't cover the entire time period?

Supplementary Fig.16: if the speleothem records on the right panel are not used in the main text and don't add valuable information for the discussion I would simply remove them.

References:

Barker, S., Diz, P., Jautravers, M. J., Pike, J., Knorr, G., Hall, I. R., and Broecker, W. S.: Interhemispheric Atlantic seesaw response during the last deglaciation, *Nature* 457, 1097–1102, <https://doi.org/10.1038/nature07770>, 2009.

Caley, T., Roche, D. M., and Renssen, H.: Orbital Asian summer monsoon dynamics revealed using an isotope enabled global climate model, *Nature Communications*, 5, 377–380, [doi:10.1038/ncomms6371](https://doi.org/10.1038/ncomms6371), 2014.

Cheng, H., Edwards, R. L., Broecker, W. S., Denton, G. H., Kong, X., Wang, Y., Zhang, R., and Wang, X.: Ice Age Terminations, *Science*, 326, 248–252, [doi:10.1126/science.1177840](https://doi.org/10.1126/science.1177840), 2009.

Pausata, F., Battisti, D. S., Nisancioglu, K. H., and Bitz, C. M.: Chinese stalagmite $\delta^{18}\text{O}$ controlled by changes in the Indian monsoon during a simulated Heinrich event, *Nat. Geosci.*, 4, 474–480, <https://doi.org/10.1038/ngeo1169>, 2011.

Severinghaus, J. P., Beaudette, R., Headly, M. A., Taylor, K., and Brook, E. J.: Oxygen-18 of O_2 records the impact of abrupt climate change on the terrestrial biosphere, *Science*, 324, 1431–1434, [doi:10.1126/science.1169473](https://doi.org/10.1126/science.1169473), 2009.

Sun, Y., Clemens, S. C., Morrill, C., Lin, X., Wang, X., and An, Z.: Influence of Atlantic meridional overturning circulation on the East Asian winter monsoon, *Nat. Geosci.*, 5, 46–49, <https://doi.org/10.1038/ngeo1326>, 2012.

Wang, Y. J., Cheng, H., Edwards, R. L., Kong, X., Shao, X., Chen, S., Wu, J., Jiang, X., Wang, X., and An, Z.: Millennial- and orbital-scale changes in the East Asian monsoon over the past 224,000 years, *Nature*, 451, 1090–1093, <https://doi.org/10.1038/nature06692>, 2008.

Reviewer #2 (Remarks to the Author):

A very brief summary of the paper:

- The authors would like to increase our understanding of climate dynamics during millennial-scale events during different boundary conditions.
- In this paper, they examine the sequence of climatic events linked to Heinrich Event 2 during the last glacial period.
- For this, they develop climate records from speleothems from the Asian monsoon and the South American monsoon domains. This provides absolute age control while using speleothem oxygen isotopic records as a proxy to track changes in climate conditions. The excellent age control also allows exploration of leads and lags thus providing further information on climate dynamics.
- The absolute age control afforded by speleothems is further used to correct the Greenland and

Antarctica ice core chronologies for this time period. For this, one speleothem record from Cherrapunji India has been used. This speleothem shows distinct annual laminae when viewed using confocal microscopy providing further control on the growth period.

- Ice core dust records, shown to be sourced dominantly from the Asian monsoon region during this time period by other referenced publications, are used to tie the ice core records to the Cherrapunji speleothem record from India.

Overall comments:

- I think the goal of the paper is vital, timely and of wide interest.

- The authors make the most of the available techniques, samples, previously published results and our current understanding of climate mechanisms.

- Given that the paper uses speleothem-based techniques to suggest a correction of ice core records, and provides a timing and sequence of events in three monsoon regions (South American, Indian and East Asian)... I give a few suggestions below that may help improve our understanding of the uncertainty on the Cherrapunji speleothem record and make the information from the individual monsoon regions more accessible.

- I am less familiar with ice core records and I hope other reviewers would be able to provide more understanding of results based on that archive.

Cherrapunji speleothem record (Figure 1; Supplementary figures 3, 4 and 5):

- The Cherrapunji speleothem record is based on 2 samples, Cherrapunji-2 and Cherrapunji-2017-1.

- Cherrapunji-2 is a longer record but the speleothem is broken and has a hiatus. The most positive oxygen isotope excursion is located close to this break and hiatus. This is not replicated by Cherrapunji-2017-1.

- The choice of one speleothem versus the other does not impact the timing of the event but effects the structure of the event, the characterisation of AHP2 I, II and III and any comparisons and discussions based on the structure of the event.

- I was wondering what the authors make of this excursion? Is there a change in mineralogy or texture of the sample at the location? How do the carbon isotopic records of the two speleothems compare? If these measurements have been made, it would be helpful to see the results. Are there other millennial scale events from these two speleothems or speleothems in close-by caves? What is the magnitude of the $\delta^{18}\text{O}$ excursion for other similar events?

- I would be wary of choosing one speleothem over the other without more explanation for why the records differ. If the authors choose to use only 1 of the 2 speleothems, it would be nice to have more convincing reasons for this. If not, the authors could still show both speleothem records without compositing them, and then being more tentative with their interpretations based on the structure of the event.

- Figure 1 for example would perhaps change with Cherrapunji being similar to the other Asian Monsoon record i.e. Yongxing record in structure without the sharp rebound at the end of III. And the South American records from Marota, Pixao and Botuvera caves having similar structures with a rebound at the end of III. Perhaps the difference is between hemispheres rather than tropics versus mid-latitudes.

- I would be curious to see if Figure 4 for example would also change if the authors similarly considered all 3 speleothems from the Asian monsoon domain. The Diamante cave record has very low resolution, perhaps the authors could show the marker points for this one.

- With the current figures, it is difficult to make out how the U-Th ages compare with the growth period established by band counting using the confocal images. It would be handy to show a figure similar to the one made by Liu et al, 2013 Figure S3 on examining the 8.2 ky event from Heshang cave. This figure would have depth on the X axis and age on the Y axis. The U-Th ages can be shown including error. The band width measurement against the count can be shown along with the U-Th ages. Such a figure gives an idea of how the band counting compares to the U-Th ages and errors with better detail. It also shows where band counting was not possible, hiatus periods etc.

- Since these speleothems are the basis for further more impactful conclusions, it would be nice to have a figure showing the break, hiatus periods and the two different oxygen isotopic records in the main paper.

Selection of speleothem records from the different monsoon regions:

- I was surprised not to see any classic EASM record in the main analysis. Perhaps the age control or resolution are insufficient... A tool like the SISAL database which allows screening with objective selection criteria e.g. location, growth period, resolution, number of U-Th dates, mineralogy would ensure that (i) the authors are not missing any additional dataset that may provide more information for analysis and (ii) provide an objective criteria for the selection of records for such an analysis. ... it is possible that no more records will emerge from the search than the authors have already selected. The authors have also used records that are not in the database so I don't doubt that they have a much better idea of what suitable records that cover this study period. Nevertheless, mining databases is a handy way of ensuring that record selection is objective.

Correlation between Greenland ice core and Cherrapunji speleothem records:

- I am not familiar with current gaps in ice core chronology development. This comment is only regarding tie points. For the tie points in Figure 2 again, it would be good to show both records from Cherrapunji and to use a more statistically robust approach such as outlined in Rehfeld et al, 2014 'similarity estimators for irregular and age uncertain time series'. Perhaps that is the wrong statistical tool, but some tie points visually appear more convincing than others and I was wondering if there was a more objective way to consider them.

Line comments:

Line 100: Use 'weakening' and 'strengthening' versus 'weakened' and 'strengthened'.

Line 102: weakening and strengthening of the monsoon systems 'respectively'

Line 103: Needs a reference.

Line 142: causal link between what?

Line 179: I think it is difficult to make this statement without comparing more records in these regions.

Line 205: Sofular record: Neither the authors in the original publication nor here fully describe the interpretation of the carbon isotope record in the context of Asian Westerlies. Perhaps you could add a sentence to make this more clear.

Line 245: 'seldom' instead of 'seldomly'.

Line 245: Do other Heinrich events line up with ice core dust records?

Line 256: Re DO events generally muted in tropical speleothem $\delta^{18}\text{O}$ records?

Line 319: What could cause the rebound in tropical regions?

Line 417: 'resumption' instead of resume.

Line 431: 'described in detail' instead of detailly.

Line 436: Which precipitation dipole?

Line 471: Are there visible layers? Which season do you think gives rise to the pulse of fluorescence material?

Supplementary section 1.3: Higher resolution measurements of the same sample compared to prior measurements at lower resolution is not a robust replication test. And replication of multiple speleothems from the same cave/region suggests a common strong driver of the oxygen isotope signal but the absolute oxygen isotope values may or may not be in equilibrium.

Supplementary section 1.5.2: Expand 'ADS'

Supplementary section 1.5.2: Westerlies 'were' presumably stronger than today.

Supplementary section 1.7: First line 'Antarctic' word repeated.

A point-by-point response to the reviews

(Original comments are in *blue*, and our responses are in *black*)

Comments from Reviewer #1

Related changes are highlighted in green in the revised manuscript attached below our answers.

Comment 1. Dong et al. present a new study that investigates the changes in atmospheric and oceanic processes in the tropical regions and Southern Hemisphere during the HE2 millennial event. They perform new oxygen isotopic measurements on eight speleothem records from mid to low latitudes (Asia and South America) at multi-annual resolution over HE2 as well as ^{230}Th dating. Based on their speleothems records and other constraints, the authors proposed to revise the ice core chronologies of +320 y for Greenland and +400 y for Antarctica, which allows for a synchronous HE2 onset globally. They also investigate the monsoon dynamic during this period and proposed to use the Greenland Ca^{2+} concentration record as a qualitative proxy for change in mid-latitude westerly winds position and intensity in the Northern hemisphere, and hydroclimate conditions in the Asian dust source region.

The paper is well-written and complete, with many aspects addressed. The work is technically rigorous and the methodology is exhaustive, with the different sensitivity tests performed and details on the construction of the new chronologies. The supplementary information is however a bit dense for the reader with many figures. This study is interesting and worth to be published in Nature Communications. I do have small comments which I hope can help improve the manuscript.

We thank the reviewer for her/his positive evaluation of our manuscript.

Comment 2. The term “Heinrich event” is probably not appropriated for this study as it refers to the cooling of the sea surface temperature synchronous with the IRD inputs observed in the North Atlantic. Since the authors present records in the mid-low latitudes with no direct evidence of IRD deposition, I would suggest to use the term “Heinrich stadial” (in the main text and figures) that denotes the cold phase in which occurred the Heinrich event (Barker et al., 2009).

Following the comment, we have added a sentence in the revised manuscript from Line #99 to 102: “*The marine records show dramatic responses to Heinrich events for extended periods of time, often towards the end of the corresponding Greenland Stadial (e.g., Barker et al., 2009). We refer to these periods of extreme conditions as Heinrich Stadials (HSs), and note that they are not defined by the duration of the Greenland Stadial counterpart.*” We have made the changes accordingly throughout the revised text and figures.

Comment 3. The authors interpret the speleothem $\delta^{18}\text{O}$ record as a proxy for monsoon circulation in India but the interpretation of this proxy is still in debate. Some studies interpreted the speleothem $\delta^{18}\text{O}$ as a proxy for monsoon intensity (related to precipitation changes in response to external climate forcing – e.g. Wang et al., 2008; Cheng et al., 2009), whereas others suggest that the speleothem $\delta^{18}\text{O}$ should not be considered as a summer monsoonal proxy only and that it reflects changes in moisture sourcing (e.g. Pausata et al., 2011; Caley et al., 2014).

This should be clarified in the manuscript.

To clarify the issue, we have added the following sentence in the revised text from Line#161 to 165: “*Although there were different views in the early studies regarding the speleothem $\delta^{18}\text{O}$ interpretation in the Asian summer monsoon domain (e.g., Wang et al., 2008; Cheng et al., 2009; Pausata et al., 2011; Caley et al., 2014), the recent developments have shown a general consensus that the speleothem $\delta^{18}\text{O}$ variations on millennial-to-orbital timescales reflect the large-scale monsoonal circulation/rainfall, which is closely linked to the overall monsoon intensity, instead of local rainfall amount (e.g., Cheng et al., 2019, 2021-2; Kathayat et al., 2021; Sun et al., 2022) (Supplementary Text 1.3)*”.

Additionally in the supplementary Note 1.3, we have added the following text: “*The earlier studies show apparent discrepancies in the speleothem $\delta^{18}\text{O}$ interpretations in the Asian summer monsoon regions at millennial-to-orbital timescales (e.g., Wang et al., 2008; Cheng et al., 2009; Pausata et al., 2011; Caley et al., 2014). For example, Yuan et al. (2004) suggested that changes in the isotopic fractionation of water vapor along the moisture trajectory between tropical ocean sources and the cave site could explain the speleothem $\delta^{18}\text{O}$ variations. Moreover, Cheng et al. (2009) suggested that changes in the annual proportion of the lighter $\delta^{18}\text{O}$ monsoon rainfall (essentially summer rainfall) could also explain the speleothem $\delta^{18}\text{O}$ variations. Results from the numerical climate model (Pausata et al., 2011) showed that during the Heinrich events, the heavier speleothem $\delta^{18}\text{O}$ values in the East Asian summer monsoon domain reflect the upstream transportation of isotopically enriched water vapor, which also corresponds to the weakening of the Indian summer monsoon. The climate models (Pausata et al., 2011) suggest that the speleothem $\delta^{18}\text{O}$ reflects spatially integrated monsoon rainfall between tropical ocean sources and cave sites, which further confirms, rather than contradicts, the speleothem proxy interpretation of Cheng et al. (2009) and Yuan et al. (2004). The recent developments in the field of modern precipitation isotopes and climate model simulations together with the proxy studies have suggested a broad consensus that at millennial-to-orbital timescales speleothem $\delta^{18}\text{O}$ in the Asian summer monsoon domain reflect to first-order the large-scale monsoonal circulation controlled by the moisture sources and the changes driven by overall monsoonal circulation patterns, which are independent of the cave locations, and precipitation amount at the site (e.g., Cheng et al., 2016, 2019, 2021-2; Kathayat et al., 2021; Zhao et al., 2019; Sun et al., 2022).*”

Comment 4. My main concern is about the comparison between speleothem $\delta^{18}\text{O}$ records and marine records in the North Atlantic Ocean that deserves to be more developed than the few sentences in the Supplementary text 1.9 about the adaptation of the sediment cores chronology with older age by 320 y. It would be nice to make the relationship between the changes in the low latitude and in the North Atlantic (climate, oceanic circulation...) and to discuss about the improvement of the chronology for these archives. The authors discussed in the main text about AMOC changes associated with changes in sea-surface salinity following changes in precipitation but don't refer to any marine records to support these hypotheses. Water-hosing modelling experiments also exist to show the response of the monsoon wind and precipitation in East Asia following a reduction of the AMOC (e.g., Sun et al., 2011). These two aspects worth to be developed in the discussion.

As per the reviewer' suggestions, we have added the following sentences in the revised text

from Line#318 to 321: “A recent study (Waelbroeck et al., 2019) has established age models for the 92 published marine sediment records from the Atlantic Ocean via correlating to the Greenland ice core GICC05 chronology. This, in turn, allows us to simply shift the chronologies of these marine records by +320-year across the HS2 period (Supplementary Fig. 15 and Supplementary Text 1.10) to compare the marine records with the precisely-dated speleothem records.”

We also added the following sentences in the revised text in Line #332 to 334: “This is also consistent with the observation that the weakening of the AMOC coincides with the waning of the Asian summer monsoon within the marine age uncertainties (Supplementary Fig. 15), similar to the results of water-hosing experiments (e.g., Sun et al., 2012).”

Regarding the salinity, in the revised version we have added the following text from Line #427 to 429: “The sea-surface salinity reconstructed from the eastern subpolar North Atlantic (MD95-2002 Core) indeed exhibits an increasing trend during the AHP2 termination (Supplementary Fig. 15)”. In addition, as mentioned in the caption of Supplementary Fig. 2, the available marine records around the Northeast Brazil show a decrease of rainfall during the SAHP2 termination (or increase of the salinity), but these marine records have approximately ± 300 –700 years uncertainties which eliminate the precise correlation with the speleothem records.

Comment 5. Other proxies of the low latitude hydrological cycle exist like the $\delta^{18}\text{O}_{\text{atm}}$ (at high resolution in Siple Dome ice core over the time period considered in this manuscript – Severinghaus et al., 2009). To support the active role of the low latitude hydroclimate dynamics during millennial events in the discussion, a comparison between the speleothem records and the $\delta^{18}\text{O}_{\text{atm}}$ could be added since positive anomalies are observed during Heinrich stadials, following a shift of the ITCZ.

This is a good suggestion. We updated the previous Fig. 3 (now Fig. 4) by adding the composite Δ_{ELAND} proxy (global terrestrial biosphere) inferred by the $\delta^{18}\text{O}_{\text{atm}}$ record (Seltzer et al., 2017). It is noted that the Δ_{ELAND} record is on the WD2014 chronology and we thus, shifted it by +400 years based on our correlations. We also added the following text in Line #330 to 332: “A positive anomaly can also be observed in the Δ_{ELAND} record (Fig. 4d), which also supports the southward shift of the tropical rainfall and terrestrial oxygen production during the HS2 (Seltzer et al., 2017; Severinghaus et al., 2009)”.

Comment 6. Specific comments:

Line 102: precise “of the two monsoon systems”.

Done.

Line 105: instead of “muted” I would rather say “relatively small”.

Done.

Line 130: an introduction of the AHP2 and SAHP2 is needed.

According to the suggestion, we have added the text in the revised version from Line #133 to 135: “AHP2, SAHP2 and HS2 are typical millennial-scale events occurred during the LGM (Supplementary Fig. 1)^{3,26}, and it remains an important task to rigorously test the climate signal propagations and phase relationships between the events”.

Line 135: what about speleothem $\delta^{18}\text{O}$ records from South American monsoon domain?

Speleothem records from the South American summer monsoon domain have been added from Line #146 to 147: “Speleothem $\delta^{18}\text{O}$ records from the South American summer monsoon domain are characterized by sub-centennial age-model uncertainties (<80 years) at the key intervals”.

Line 151: Fig. 1B and 1C.

Done.

Line 160: precise the resolution of the measurements.

We have included the following text from Line #174 to 175: “The spatial-resolution of the measurements varies between 0.05 and 1 mm (see Methods)”.

I also wouldn't say that there is an “excellent replication between $\delta^{18}\text{O}$ records from the same cave” since there is a shift in the absolute $\delta^{18}\text{O}$ value between previous and new measurements for Yongxing and Paixão caves, even if they both show the same variations (see comment on Supplementary text 1.3).

We agree with the reviewer. We have rephrased the sentence from Line #175 to 177: “The comparison between the $\delta^{18}\text{O}$ records from the same and different caves in the same climatic region (Supplementary Fig. 6) suggests that the speleothem $\delta^{18}\text{O}$ records broadly replicate although there are minor differences in their absolute values”.

Line 236: either the z-score value is >-1 or a tie point is missing on Supplementary Fig. 12 to show the tie point with a z-score value >4.

We have rephrased the sentence as follows (Line #257 to 258): “Another tie point is at the start of AHP2, from which the Cherrapunji $\delta^{18}\text{O}$ increase abruptly to the high z-score values (>4) (Supplementary Fig. 13)”. Additionally, we highlighted the abrupt transition in Supplementary Fig. 13 using an arrow.

Line 318: delete one “monsoon” term.

Done.

Lines 368-369: is it possible to state on the role of the ITCZ position on the CH_4 and to decipher the role of low latitude wetlands vs high latitude emissions?

Following the comment, we have rephrased the sentence to focus on the concurrence between the abrupt change in the SASM domain $\delta^{18}\text{O}$ and ice-core CH_4 on the improved chronology in Line #399 to 401: “This common signal provides compelling evidence that enhanced rainfall in the Southern Hemisphere tropical wetlands indeed coincides with a significant CH_4 excess early in AHP2, favoring the previous hypothesis”.

Line 462: precise that the $\delta^{13}\text{C}$ measurements are not shown in this paper.

We have added the $\delta^{13}\text{C}$ plot in supplementary Fig. 4c and the data in Supplementary Data 2. However, the climatic significance of Cherrapunji $\delta^{13}\text{C}$ data is complex, which requires further study in the future.

Lines 523-524: it is not clearly explained how the search time windows used for the Ramp-fitting technique in Supplementary Fig.7 and 9 are defined. Even if the text indicates that they are randomly chosen, I guess that there are still some early constraints.

We added more information about the technique in the text from Line #565 to 567: *“The main criteria to choose analytical time intervals are as follows: 1) the interval contains two prominent change points for Ramp-fitting, and 2) the same time intervals are used for records from the same region if possible.”*

Supplementary text 1.2: why the counting for Cherrapunji-2017-1 is manual and not machine-assisted?

The main purpose of using the machine-assisted method is to obtain the thickness data of annual bands. The annual lamina thickness data of the sample Cherrapunji-2 have been added in the new Fig. 1 and Supplementary Fig. 4. However, the annual laminae of Cherrapunji-2017-1 are much thinner, and thus difficult to be measured by using the current machine-assisted method. Therefore, we only estimate the annual growth rate for the sample Cherrapunji-2017-1. In the revised Supplementary Note 1.2, we added the following text: *“The annual lamina thickness of Cherrapunji-2 obtained by the machine-assisted method is shown in Fig. 1b and Supplementary Fig. 4a”*.

Supplementary text 1.3: previous and new \$\delta^{18}\text{O}\$ measurements for Yongxing and Paixão caves have been done on the same speleothem. So when comparing the previous and new measurements I don't expect a large shift between the two, as observed in Supplementary Fig. 5a between 24.5–24 ka and in Supplementary Fig. 5c between 24.5–25 ka. This need to be explained (analysis not done in the same laboratory, with different protocole/instrument, fractionation effect...).

The new $\delta^{18}\text{O}$ data of samples YX-51 and PX-07 show slightly heavier values (0.5‰ for YX-51 and 0.8‰ for PX-07) compared to the previous measurements, although the structures are similar. The differences in the absolute $\delta^{18}\text{O}$ value might be due to the different analysis protocols/instrumentations between different laboratories. We added this information in the revised manuscript.

Supplementary text 1.5.2: “...during the ADS dust season...” please rephrase.

Changed to *“during the dust season in the Asian dust source regions”*.

In the sentence “between dust emission in ADS region”, correct the word “emission”.

Done.

Supplementary text 1.5.3: is there a reason for similar Greenland \$\text{Ca}^{2+}\$ concentration with

smaller amplitude in $\delta^{18}\text{O}$ during 26.5–25.3 ka than during AHP2? Could the Greenland record not only be linked to changes in the Asian dust source (for example with another influence from the Northern Hemisphere)?

Good question. We added the following discussion in the revised Note 1.6.3: “Based on this observation, we surmise that although the “stadial” between 26.5–25.3 ky BP is the non-Heinrich type, its climatic impact on the meridional position/strength of the Northern Hemisphere westerly wind might be comparable to the Heinrich stadial 2. Alternatively, this may suggest an influence from other potential dust sources besides the Asian dust sources, which warrants further study”.

Supplementary text 1.6: I would also have considered an additional uncertainty₄ associated with the ^{230}Th dating method to obtain the Cherrapunji record chronology.

This uncertainty has already been factored. The tie point uncertainty in the Cherrapunji record includes uncertainties associated with both the age model and change point estimation. The former is basically from the ^{230}Th dating errors to obtain the Cherrapunji record chronology. To clarify, we have added the following information in the Supplementary Note 1.7: “The tie point uncertainty of the Cherrapunji record includes both age model uncertainty and change point uncertainty (Supplementary Table 2).

Supplementary text 1.7: explain that the d_{in} is the logarithmic definition of deuterium excess that better preserves isotopic moisture source information than the linear definition.

Done.

In the sentence “...it is commonly used as a proxy to trace...”, remove “as a proxy”.

Done.

Supplementary Fig.2: precise the range of “large uncertainties” of the chronology of marine records.

Done.

Supplementary Fig.10: correct the caption to “Monthly data are from...”.

Done.

The ITCZ position could also be added for both months.

Done.

Supplementary Fig.11: the $\delta^{13}\text{C}$ has been measured in the speleothems of this study. Why not showing them here instead of the So-1 record that doesn't cover the entire time period?

The previous Supplementary Fig. 11 (now supplementary Fig. 12) aims to show the proxy records dominantly influenced by the Asian westerlies. Therefore, in this figure we included the So-1 $\delta^{13}\text{C}$ record from the Asian Westerlies domain. We avoided our Cherrapunji record in this comparison because Cherrapunji Cave is located at the Indian summer monsoon domain,

but we have now shown the Cherrapunji $\delta^{13}\text{C}$ profile in the Supplementary Fig. 4c. In the revised version of the manuscript, we have added a clarification from Line #228 to 230: “During the AHP2 onset, the So-1 $\delta^{13}\text{C}$ record exhibits an abrupt enrichment, indicating a change to the colder/drier condition (Fleitmann et al., 2009), in line with the southward shift of the Westerlies”.

Supplementary Fig.16: if the speleothem records on the right panel are not used in the main text and don't add valuable information for the discussion I would simply remove them.

Done.

References:

- Barker, S., Diz, P., Jautravers, M. J., Pike, J., Knorr, G., Hall, I. R., and Broecker, W. S.: Interhemispheric Atlantic seesaw response during the last deglaciation, *Nature* 457, 1097–1102, <https://doi.org/10.1038/nature07770>, 2009.
- Caley, T., Roche, D. M., and Renssen, H.: Orbital Asian summer monsoon dynamics revealed using an isotope enabled global climate model, *Nature Communications*, 5, 377–380, doi:10.1038/ncomms6371, 2014.
- Cheng, H., Edwards, R. L., Broecker, W. S., Denton, G. H., Kong, X., Wang, Y., Zhang, R., and Wang, X.: Ice Age Terminations, *Science*, 326, 248–252, doi:10.1126/science.1177840, 2009.
- Pausata, F., Battisti, D. S., Nisancioglu, K. H., and Bitz, C. M.: Chinese stalagmite $\delta^{18}\text{O}$ controlled by changes in the Indian monsoon during a simulated Heinrich event, *Nat. Geosci.*, 4, 474–480, <https://doi.org/10.1038/ngeo1169>, 2011.
- Severinghaus, J. P., Beaudette, R., Headly, M. A., Taylor, K., and Brook, E. J.: Oxygen-18 of O_2 records the impact of abrupt climate change on the terrestrial biosphere, *Science*, 324, 1431–1434, doi:10.1126/science.1169473, 2009.
- Sun, Y., Clemens, S. C., Morrill, C., Lin, X., Wang, X., and An, Z.: Influence of Atlantic meridional overturning circulation on the East Asian winter monsoon, *Nat. Geosci.*, 5, 46–49, <https://doi.org/10.1038/ngeo1326>, 2012.
- Wang, Y. J., Cheng, H., Edwards, R. L., Kong, X., Shao, X., Chen, S., Wu, J., Jiang, X., Wang, X., and An, Z.: Millennial- and orbital-scale changes in the East Asian monsoon over the past 224,000 years, *Nature*, 451, 1090–1093, <https://doi.org/10.1038/nature06692>, 2008.

Comments from Reviewer #2

Related changes are highlighted in yellow in the revised manuscript attached below our answers.

Comment 1. A very brief summary of the paper:

- The authors would like to increase our understanding of climate dynamics during millennial-scale events during different boundary conditions.
- In this paper, they examine the sequence of climatic events linked to Heinrich Event 2 during the last glacial period.
- For this, they develop climate records from speleothems from the Asian monsoon and the

South American monsoon domains. This provides absolute age control while using speleothem oxygen isotopic records as a proxy to track changes in climate conditions. The excellent age control also allows exploration of leads and lags thus providing further information on climate dynamics.

- The absolute age control afforded by speleothems is further used to correct the Greenland and Antarctica ice core chronologies for this time period. For this, one speleothem record from Cherrapunji India has been used. This speleothem shows distinct annual laminae when viewed using confocal microscopy providing further control on the growth period.

- Ice core dust records, shown to be sourced dominantly from the Asian monsoon region during this time period by other referenced publications, are used to tie the ice core records to the Cherrapunji speleothem record from India.

Overall comments:

- I think the goal of the paper is vital, timely and of wide interest.

- The authors make the most of the available techniques, samples, previously published results and our current understanding of climate mechanisms.

- Given that the paper uses speleothem-based techniques to suggest a correction of ice core records, and provides a timing and sequence of events in three monsoon regions (South American, Indian and East Asian)... I give a few suggestions below that may help improve our understanding of the uncertainty on the Cherrapunji speleothem record and make the information from the individual monsoon regions more accessible.

- I am less familiar with ice core records and I hope other reviewers would be able to provide more understanding of results based on that archive.

We thank the reviewer for her/his positive evaluation of our manuscript.

Comment 2. Cherrapunji speleothem record (Figure 1; Supplementary figures 3, 4 and 5):

- The Cherrapunji speleothem record is based on 2 samples, Cherrapunji-2 and Cherrapunji-2017-1.

- Cherrapunji-2 is a longer record but the speleothem is broken and has a hiatus. The most positive oxygen isotope excursion is located close to this break and hiatus. This is not replicated by Cherrapunji-2017-1.

- The choice of one speleothem versus the other does not impact the timing of the event but effects the structure of the event, the characterisation of AHP2 I, II and III and any comparisons and discussions based on the structure of the event.

- I was wondering what the authors make of this excursion? Is there a change in mineralogy or texture of the sample at the location? How do the carbon isotopic records of the two speleothems compare? If these measurements have been made, it would be helpful to see the results. Are there other millennial scale events from these two speleothems or speleothems in close-by caves? What is the magnitude of the $\delta^{18}\text{O}$ excursion for other similar events?

Very good questions.

(1) As shown in Figure R1, there is no observed changes in terms of mineralogy or texture at the heaviest $\delta^{18}\text{O}$ excursion, compared with the upper section.

Figure R1. The image of the section of the sample Cherrapunji-2 around the $\delta^{18}\text{O}$ large excursion (the red dashed rectangle). The orange arrow shows the hiatus position.

(2) The annual laminae are continuous across the $\delta^{18}\text{O}$ large excursion (Supplementary Fig. 3b or Figure R2 below).

Figure R2. The confocal microscopy image of the section of the sample Cherrapunji-2 across the $\delta^{18}\text{O}$ large excursion. Red curve is the plot of Cherrapunji-2 $\delta^{18}\text{O}$ versus depth. The ages of change points and the age model uncertainty (2σ) are marked (red arrows).

(3) In the updated version, the carbon isotopic profiles of both the speleothems are shown in the supplementary Fig. 4c. Both the Cherrapunji-2017-1 and Cherrapunji-2 $\delta^{13}\text{C}$ records show similar abrupt positive excursions at the AHP2 onset. However, the interpretation of Cherrapunji $\delta^{13}\text{C}$ records remains complex and further studies are needed to understand the $\delta^{13}\text{C}$ data.

(4) Apart from HS2 there are a few published Heinrich Stadial (HS1 and HS4) records from Northeast India (e.g., Dutt et al., 2015, Doi: 10.1002/2015GL064015; Cheng et al., 2021, Doi: 10.1038/s43247-021-00304-6). The detailed characterization (structure, timing and duration)

of each H-event is different due to different boundary conditions (Supplementary Fig. 1).

Comment 3. I would be wary of choosing one speleothem over the other without more explanation for why the records differ. If the authors choose to use only 1 of the 2 speleothems, it would be nice to have more convincing reasons for this. If not, the authors could still show both speleothem records without compositing them, and then being more tentative with their interpretations based on the structure of the event.

Good suggestion. Here we provide more information about our reasoning.

(1) As shown in the revised Figure 1 (or Figure R3 below), the $\delta^{18}\text{O}$ of the slower growth speleothem Cherrapunji-2017-1 is systematically heavier by 0.6‰ than the faster growth Cherrapunji-2, thus the Cherrapunji-2017-1 record is adjusted accordingly. Noticeably, a slight offset remains observable between 24.3–24.0 ky BP. This difference stems from the heavier $\delta^{18}\text{O}$ of Cherrapunji-2017-1 accompanied with its extremely slow growth period (~0.009 mm/year) (Figure 1 or Figure R3 below), which is commonly observed in many caves (e.g., Wang et al., 2001; Stoll et al., 2015; Tan et al., 2019).

Figure R3. Cherrapunji Cave speleothem records. (a) Speleothem $\delta^{18}\text{O}$ records from Cherrapunji Cave (Cherrapunji-2 and Cherrapunji-2017-1, this study) were used to construct a composite record. Over their contemporary growth intervals, we exclusively use the Cherrapunji-2 record (Supplementary Note 1.1). Error bars show ^{230}Th dates with uncertainties (2σ) for each record (color coded). (b) Annual lamina thickness of speleothem Cherrapunji-2 and estimated annual growth rate of speleothem Cherrapunji-2017-1. The dotted black box shows the period with extremely slow growth rate in Cherrapunji-2017-1, and the corresponding $\delta^{18}\text{O}$ data in (a) is depicted by dots.

(2) Regarding the chronology, the chronological constraints and age precision of speleothem Cherrapunji-2 are comparatively robust. The interval between 25.4–23.7 ka BP in Cherrapunji-2 is constrained by 17 ^{230}Th dates whereas Cherrapunji-2017-1 has only 10 ^{230}Th dates (Figure R3).

(3) The overall amplitude and structure of the AHP2 is replicated by the speleothem MWS-1 from nearby Mawmluh cave (Supplementary Fig. 6d or Figure R4 below).

Figure R4. The replication between the MWS-1 (Mawmluh cave) and Cherrapunji composite records in terms of their amplitudes and structure.

(4) In the revised manuscript, we added a new speleothem record (DQH-17) from Dongqinghe cave ($32^{\circ} 34' \text{ N}$, $106^{\circ} 12' \text{ E}$). This cave and Wulu cave (Zhao et al., 2010) are located in the transitional zone between ISM and EASM domains (Wang and Lin, 2002. Doi: [https://doi.org/10.1175/1520-0442\(2002\)015<0386:RSOTAP>2.0.CO;2](https://doi.org/10.1175/1520-0442(2002)015<0386:RSOTAP>2.0.CO;2)) (Figure R5), so that one would expect to see the heaviest $\delta^{18}\text{O}$ excursion at ~ 24.3 ky BP to some extent similar to the Cherrapunji-2 record. Indeed, although the amplitudes are different as expected (e.g., Pausata et al., 2011, Doi: 10.1038/NGEO1169), both DQH-17 from Dongqinghe cave and Wu-32 from Wulu cave records show the heaviest $\delta^{18}\text{O}$ excursion at ~ 24.3 ky BP, similar with the Cherrapunji record, providing a robust replication in terms of the structure (Figure R5 or Supplementary Fig. 7).

In sum, above lines of evidence suggest that our Cherrapunji composite record is robust in terms of the structure and chronology.

Figure R5. Comparison between EASM and ISM records across the AHP2. Upper panel: the location map of Cherrapunji cave site in the ISM domain, Dongqinghe and Wulu cave sites in the transitional zone between ISM-EASM domains, and Yongxing and Hulu cave sites in the EASM domain. Lower panel: (a) to (e) Hulu, Yongxing, Dongqinghe, Wulu and Cherrapunji cave $\delta^{18}\text{O}$ records. Both Dongqinghe and Wulu cave records show the heaviest $\delta^{18}\text{O}$ excursion at ~ 24.3 ky BP, similar to the Cherrapunji record, providing a robust replication in terms of this structure, while the Hulu and Yongxing records do not have such a structure. Error bars in (b), (d) and (e) show ^{230}Th dates with uncertainties (2σ) for each record. The error bars in (a) and (c) depict the age model uncertainties of the Hulu Cave record (based on Cheng et al. (2018)) and Dongqinghe Cave record (this study). The ISM and EASM trajectories follow Cheng et al. (2012).

Comment 4. - Figure 1 for example would perhaps change with Cherrapunji being similar to the other Asian Monsoon record i.e. Yongxing record in structure without the sharp rebound at the end of III. And the South American records from Marota, Pixao and Botuvera caves having similar structures with a rebound at the end of III. Perhaps the difference is between hemispheres rather than tropics versus mid-latitudes.

The PA-LA-1 record from the tropical American monsoon regime (in Northern Hemisphere) also shows a sharp rebound structure similar to the Cherrapunji record, which we have highlighted in the updated version of Fig. 5. The detailed discussion about the proxy records is in the revised version of the manuscript from line # 346 to 354, supplementary Note 1.9 and also provided in our answers to the comments 3.

Comment 5. - I would be curious to see if Figure 4 for example would also change if the authors similarly considered all 3 speleothems from the Asian monsoon domain. The Diamante cave record has very low resolution, perhaps the authors could show the marker points for this one.

In the updated version, we have added YX-51 data in the Fig. 5. The marker points for the Diamante Cave record are also shown in the Fig. 6. The detailed discussion is provided in our answers to the comments 3.

Comment 6. - With the current figures, it is difficult to make out how the U-Th ages compare with the growth period established by band counting using the confocal images. It would be handy to show a figure similar to the one made by Liu et al, 2013 Figure S3 on examining the 8.2 ky event from Heshang cave. This figure would have depth on the X axis and age on the Y axis. The U-Th ages can be shown including error. The band width measurement against the count can be shown along with the U-Th ages. Such a figure gives an idea of how the band counting compares to the U-Th ages and errors with better detail. It also shows where band counting was not possible, hiatus periods etc.

We have added the figure as suggested (Supplementary Fig. 4).

Comment 7. - Since these speleothems are the basis for further more impactful conclusions, it would be nice to have a figure showing the break, hiatus periods and the two different oxygen isotopic records in the main paper.

Added in the new Fig. 1.

Comment 8. Selection of speleothem records from the different monsoon regions:

- I was surprised not to see any classic EASM record in the main analysis. Perhaps the age control or resolution are insufficient... A tool like the SISAL database which allows screening with objective selection criteria e.g. location, growth period, resolution, number of U-Th dates, mineralogy would ensure that (i) the authors are not missing any additional dataset that may provide more information for analysis and (ii) provide an objective criteria for the selection of records for such an analysis. ... it is possible that no more records will emerge from the search than the authors have already selected. The authors have also used records that are not in the database so I don't doubt that they have a much better idea of what suitable records that cover

this study period. Nevertheless, mining databases is a handy way of ensuring that record selection is objective.

We used the SISAL database to find speleothem records suitable for this study, and, we also included the previously-published high-resolution speleothem records that are not in the database yet. In the revised version we have also mentioned about the SISAL database clearly (line # 191 to 195): “In order to objectively select the speleothem records in the EASM domain, we used Speleothem Isotopes Synthesis and Analysis (SISAL) database (Atsawawaranunt et al., 2018; Comas-Bru et al., 2020) to choose the speleothem $\delta^{18}\text{O}$ records that meet at least one of the following criteria in that the records have either a temporal resolution better than 40 years or have sub-centennial age uncertainty. The high-quality EASM speleothem $\delta^{18}\text{O}$ records that are not in the database are also used in this study (Supplementary Fig. 7)”.

Comment 9. Correlation between Greenland ice core and Cherrapunji speleothem records:
- I am not familiar with current gaps in ice core chronology development. This comment is only regarding tie points. For the tie points in Figure 2 again, it would be good to show both records from Cherrapunji and to use a more statistically robust approach such as outlined in Rehfeld et al, 2014 ‘similarity estimators for irregular and age uncertain time series’. Perhaps that is the wrong statistical tool, but some tie points visually appear more convincing than others and I was wondering if there was a more objective way to consider them.

As suggested, we conducted an additional statistical analysis: the lagged correlation analysis used in various studies (e.g., WAIS Divide Project Members, 2015. Doi: 10.1038/nature14401). The composite Cherrapunji and Greenland ice-core [Ca^{2+}] records (on their original chronologies) were used. We first interpolated the raw dataset to an average resolution of 20 years and then extracted the low-frequency variability with a 400-year low-pass Butterworth filter, and calculated their lagged correlations (Figure R6). The lagged correlation agrees broadly with our +320-year shift of the Greenland chronology (Figure R6c).

Figure R6. Lagged correlation analysis between 20-year interpolated NGRIP [Ca^{2+}] and Cherrapunji $\delta^{18}\text{O}$ records. (a) The 20-year interpolated Cherrapunji $\delta^{18}\text{O}$ record (green) and overlain by the low-frequency variability (orange). The low-frequency variability was extracted with a 400-year low-pass Butterworth filter. **(b)** is the same as in **(a)** but for the NGRIP [Ca^{2+}] record (brown), which is on the originally published GICC05 chronology. **(c)** Lagged correlation of 400-year low-pass filtered records between NGRIP [Ca^{2+}] and Cherrapunji $\delta^{18}\text{O}$.

Comment 10. *Line comments*

Line 100: Use ‘weakening’ and ‘strengthening’ versus ‘weakening’ and ‘strengthened’.

Done.

Line 102: weaking and strengthening of the monsoon systems ‘respectively’

Done.

Line 103: Needs a reference.

Done.

Line 142: causal link between what?

Changed to “*the causal link between the high- and low-latitude climate systems in both hemispheres*”.

Line 179: I think it is difficult to make this statement without comparing more records in these regions.

Please refer to our response to the *Comment # 3*. We also rephrased the sentence to “*Notably, however, the stage III excursion is only evident in the records in the ISM domain and the transitional zone between the ISM and EASM domains, but absent in the EASM domain (Fig. 2 and Supplementary Fig. 7), as discussed in following sections*” in Line # 198 to 200.

Line 205: Sofular record: Neither the authors in the original publication nor here fully describe the interpretation of the carbon isotope record in the context of Asian Westerlies. Perhaps you could add a sentence to make this more clear.

According to the suggestion, we added the following information in Line # 228 to 230: “*During the AHP2 onset, the So-1 $\delta^{13}C$ record exhibits an abrupt enrichment, indicating a change to the colder/drier condition (Fleitmann et al., 2009), in line with the southward shift in the Westerlies*”.

Line 245: ‘seldom’ instead of ‘seldomly’.

Done.

Line 245: Does other Heinrich events line up with ice core dust records?

Yes, other millennial-scale events (e.g., Heinrich Stadial 4 and Younger Dryas) also line up precisely with the ice core dust records based on most recent studies (e.g., Cheng et al., 2020, 2021). We added more information as follows: “*Moreover, although other millennial-scale ASM events (e.g., HS4/AHP4 (Cheng et al., 2021) and Younger Dryas (Cheng et al., 2020)) apparently correlate with the ice core [Ca^{2+}] records (Supplementary Fig. 1), the dynamical mechanism underlying the correlation was not well explored in the previous studies*”.

Line 256: Re DO events generally muted in tropical speleothem $d^{18}O$ records?

The DO events might be muted in the tropical regions (e.g., Carolin et al., 2013, Doi: 10.1126/science.1233797), especially the DO-2 as a small DO event (e.g., Kindler et al., 2014, Doi: 10.5194/cp-10-887-2014; Rasmussen et al., 2014, Doi: 10.1016/j.quascirev.2014.09.007). As such, it is most likely that the DO-2 left a small imprint in the Cherrapunji $\delta^{18}O$ record as observed.

Line 319: What could cause the rebound in tropical regions?

The rebound is mechanistically associated with the ITCZ meridional shift, which is controlled largely by the interhemispheric temperature gradient (e.g., Zhang and Delworth, 2005, Doi: 10.1175/JCLI3460.1). However, the cause of it remains unclear. Here we characterize the event precisely and providing critical constraints for further modelling study.

Line 417: ‘resumption’ instead of resume.

Done.

Line 431: ‘described in detail’ instead of detailly.

Done.

Line 436: Which precipitation dipole?

We made a clarification as follows (#Line 471 to 473): “*The intra-seasonal to interannual rainfall variability over the Indian subcontinent exhibits a quasi-east-west precipitation dipole with anomalies of one sign over Northeast India and of an inverse sign over North, Northwest, and Central India (Sinha et al., 2011; Kathayat et al., 2021)*”.

Line 471: Are there visible layers? Which season do you think gives rise to the pulse of fluorescence material?

Although there are no layers to the naked eye (Figure R1), the annual laminae can be observed clearly under confocal laser scanning microscope (Supplementary Fig. 3). In general, the early summer monsoonal rainfall may give rise to the pulse of fluorescence material, due to the flushing of the organic matter accumulated during winter-spring.

Supplementary section 1.3: Higher resolution measurements of the same sample compared to prior measurements at lower resolution is not a robust replication test. And replication of multiple speleothems from the same cave/region suggests a common strong driver of the oxygen isotope signal but the absolute oxygen isotope values may or may not be in equilibrium.

We agree with the reviewer. We rephrased the sentence in the revised version (Supplementary Note 1.4) as follows: “*In summary, the comparison between the $\delta^{18}O$ records from the same and different caves in the same climatic region (Supplementary Fig. 6) suggests that the speleothem $\delta^{18}O$ records broadly replicate although there are minor differences in their absolute values*”.

Supplementary section 1.5.2: Expand ‘ADS’

Done.

Supplementary section 1.5.2: Westerlies ‘were’ presumably stronger than today.

Done.

Supplementary section 1.7: First line ‘Antarctic’ word repeated.

Done.

REVIEWERS' COMMENTS

Reviewer #1 (Remarks to the Author):

I thank the authors for the response they have provided. They considered and addressed each of the comments/suggestions in the revised manuscript. I have no more comment and recommend the publication of the manuscript.

Reviewer #2 (Remarks to the Author):

I am satisfied with the detailed review comments provided by the authors. They have made suitable changes to the manuscript to reflect the additional feedback provided by reviewers. I think it is a great piece of work and congratulate the authors.

A point-by-point response to the reviews

*(Original comments are in **blue**, and our responses are in **black**)*

Comments from Reviewer #1

I thank the authors for the response they have provided. They considered and addressed each of the comments/suggestions in the revised manuscript. I have no more comment and recommend the publication of the manuscript.

We thank the reviewer for her/his positive evaluation of our manuscript.

Comments from Reviewer #2

I am satisfied with the detailed review comments provided by the authors. They have made suitable changes to the manuscript to reflect the additional feedback provided by reviewers. I think it is a great piece of work and congratulate the authors.

We thank the reviewer for her/his positive evaluation of our manuscript.